# Leveraging Task Structures for Improved Identifiability in Neural Network Representations

## Abstract

This work extends the theory of identifiability in supervised learning by considering the consequences of having access to a distribution of tasks. In such cases, we show that linear identifiability is achievable in the general multi-task regression setting. Furthermore, we show that the existence of a task distribution which defines a conditional prior over latent factors reduces the equivalence class for identifiability to permutations and scaling, a much stronger and more useful result than linear identifiability. When we further assume a causal structure over these tasks, our approach enables simple maximum marginal likelihood optimization together with downstream applicability to causal representation learning. Empirically, we validate that our model outperforms more general unsupervised models in recovering canonical representations for both synthetic and real-world molecular data.

## 1 Introduction

Multi-task regression is a common problem in machine learning, which naturally arises in many scientific applications such as molecular property prediction (Stanley et al., 2021; Chen et al., 2023). Despite this, most deep learning approaches to this problem attempt to model the relationships between tasks through heuristic approaches, such as fitting a shared neural network in an attempt to capture the joint structures between tasks. Beyond lacking a principled approach to modeling task relationships, these approaches fail to account for how we may expect the latent factors for related tasks to change. In this work, we show that by leveraging assumptions about the relationships between the latent factors of the data *across* tasks, in particular that they vary sparsely in their causal and spurious relationships with the target variables, we can achieve identifiability of the latent factors up to permutations and scaling, and simultaneously identify the causal and spurious latent factors with respect to the target variable.

A common assumption in the causal representation learning literature, known as the sparse mechanism shift hypothesis (Schölkopf, 2019; Schölkopf et al., 2021; Perry et al., 2022), states that changes across tasks arise from sparse changes in the underlying causal mechanisms. While we do not operate directly on structural causal models, our result arises by similarly considering the implications of sparse changes in the causal graph defining a multi-task learning setting. We accomplish this by first extending the theory of identifiability in supervised learning to the multi-task regression setting for identifiability up to linear transformations (i.e., weak identifiability). We then propose a new approach to identifying neural network representations up to permutations and scaling (i.e., strictly strong identifiability), by leveraging the causal structures of the underlying latent factors for each task. We empirically validate our model's ability to recover the ground-truth latent structure of the data both in simulated settings where data is generated from our model and for real-world molecular data. This contrasts with current state-of-the-art approaches such as (Khemakhem et al., 2020a; Lu et al., 2022), whose assumptions also fit our assumed data generating process but which are difficult to train effectively and only identifiable up to block permutations and scaling.

## 2 BACKGROUND AND RELATED WORK

The notion of optimizing for disentangled representations gained traction in the recent unsupervised deep learning literature when it was proposed that this objective may be sufficient to improve desirable attributes such as interpretability, robustness, and generalization (Bengio et al., 2013; Higgins et al., 2017; Chen et al., 2016). However, the notion of disentanglement alone is not intrinsically well-defined, as there may be many disentangled representations of the data which are seemingly equally valid. Thus it is not clear a priori that this criterion is sufficient to achieve the above desiderata (Locatello et al., 2019). In the identifiable representation learning literature, the *correct* disentangled representation is assumed to be the one which corresponds to the ground-truth data generating process. Thus, what is required is an *identifiable* representation, which must be equivalent to the causal one for sufficiently expressive model classes. In the linear case, identifiability results exist in the classical literature for ICA, which requires non-gaussianity assumptions on the sources for the data (Herault & Jutten, 1986; Comon, 1994).

Many extensions of ICA to the non-linear case have been proposed, together with significant theoretical advances. In particular, Hyvarinen et al. (2019) extend this by assuming a conditionally factorized prior over the latent factors given some observed auxiliary variables, and propose a contrastive learning objective for recovering the inverse of the function which generated the observations. iVAE (Khemakhem et al., 2020a) further extends this to the setting of noisy observations, drawing connections with variational autoencoders (Kingma & Welling, 2013) and enabling direct optimization via a variational objective. Lachapelle et al. (2022) demonstrate that strong identifiability results remain achievable under weaker conditions on the sufficient statistics of the prior *if* the data generating process implies that the latent factors are governed by sparse mechanism shifts. iCaRL (Lu et al., 2022) derives analogous results for the case where the prior over the latent factors is a more general non-factorized exponential family distribution. However, the complex nature of the prior requires score matching, which is difficult to optimize in practice. Hälvä et al. (2021); Morioka et al. (2021) both obtain strong identifiability results by exploiting specific temporal or spatial structure in the encoded latents and modelling the joint as dynamical systems, however their models do not translate well to the static setting, and their identifiability results remain restricted to non-linear coordinate-wise transformations of the latent variables.

While these works are generally concerned with the unsupervised and semi-supervised setting, Khemakhem et al. (2020b); Roeder et al. (2021) discuss the identifiability properties of learned representations in the case of single-task supervised classification, showing that the representations obtained via the final hidden layer of a neural network are identifiable up to *linear* transformations, which may not be a sufficiently restrictive equivalence class for practical applicability. While Hyvärinen & Pajunen (1999); Khemakhem et al. (2020a) show that identifiability is not achievable without any form of conditioning in the prior, Willetts & Paige (2021); Kivva et al. (2022) recently extend the results in unsupervised generative models to the case of models with mixture model priors. This can be seen as providing analogous identifiability results to prior work, without assumptions on the observability or the dimensionality of the conditioning variable. Nonetheless, these results do not apply to the exact likelihood, and it remains unclear to what extent the practical consistency and identifiability is achievable when optimizing a surrogate objective.

In contrast, Brady et al. (2023) discuss identifiability results which arise from assumptions on the structure of the mixing function, specifically targeting dual objectives of compositionality with respect to partitions of the latent factors and invertibility of the mixing function. Thus, no distributional assumptions are made on the prior. While this approach has similarities with our proposal by introducing assumptions on how partitions of the latent space evolve with respect to well-defined objects, we propose a general setting which is not restricted to representation learning in visual scenes. Furthermore, by formalizing these assumptions within our probabilistic model, we eliminate the need for auxiliary terms in our optimization objective.

Furthermore, concurrent work (Lachapelle et al., 2023; Fumero et al., 2023) has expanded this area of research to consider the multi-task and meta-learning settings. However, their approach to achieving permutation-identifiable representations relies on introducing heuristic sparsity constraints, such as entropy and $L_2$-norm regularizers, within a bi-level optimization objective, which turns out to be difficult to solve both in theory and in practice (Sinha et al., 2017). This contrasts with the straightforward and principled optimization objective arising from our probabilistic model.

Finally, while many recent works have shown that spurious correlations are a failure case of deep learning and focus on eliminating them (Rojas-Carulla et al., 2018; Arjovsky et al., 2019; Krueger

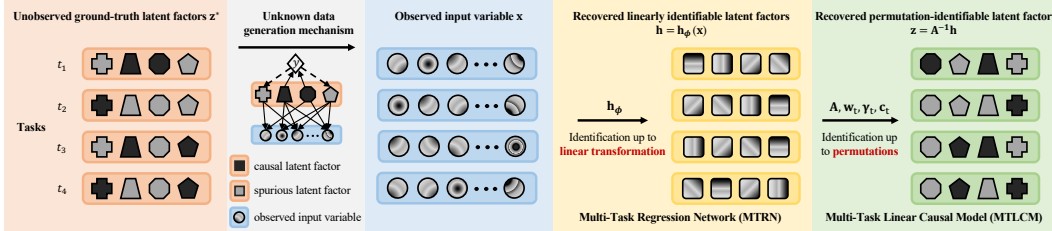

Figure 2: The workflow of our proposed method. Shapes are used to track the positions of the ground-truth and recovered latent factors. Colors are used to differentiate between causal and spurious latent factors. We assume that the observed variable is obtained by transforming the ground-truth latent factors with some mixing function. We show that a multi-task regression network (MTRN) can recover the ground-truth latent factors (i.e., data representations) up to linear transformation and further propose a multi-task linear causal model (MTLCM) to reduce the equivalence class for identifiability to permutations and scaling.

et al., 2021; Eastwood et al., 2022; Lu et al., 2022; Kirichenko et al., 2023), we leverage spurious features to *improve* the robustness of the learned representations in the multi-task setting.

## 3 PROPOSED METHOD

This section proposes a novel method that leverages task structures in the multi-task regression setting to identify the ground-truth data representations up to permutations and scaling.

### 3.1 PROBLEM FORMULATION AND ASSUMPTIONS

The assumptions of the ground-truth data generating process considered in this paper are encapsulated in the causal graph shown in Figure 1, where the input variable $\mathbf{x} \in \mathcal{X} \subseteq \mathbb{R}^n$, the target variable[1] $y \in \mathbb{R}$ and the task index variable $t \in \{1, \cdots, N_t\}$ are observed variables, and the latent factors $\mathbf{z} \in \mathbb{R}^d \ (d \le n)$ are unobserved variables. We assume that $\mathbf{x}$ is generated by transforming some (unobserved) ground-truth independent latent factors $\mathbf{z}^*$ with some unknown bijective mixing function $\mathbf{f}_* : \mathbb{R}^d \to \mathcal{X}$, i.e., $\mathbf{x} = \mathbf{f}_*(\mathbf{z}^*)$. To incorporate the sparse mechanism shift hypothesis across tasks, we further assume that each task $t$ has its own partition of the ground-truth latent factors $\mathbf{z}^* = \mathbf{z}_c^* \cup \mathbf{z}_s^*$ into a set of causal latent factors $\mathbf{z}_c^*$ and a set of spurious latent factors $\mathbf{z}_s^*$, and such partitions potentially vary across tasks. The target variable is assumed to be a weighted sum of the causal latent factors, i.e., $y = (\mathbf{w}_t^*)^{\mathrm{T}} \mathbf{z}^*$, where $\mathbf{w}_t^* \in \mathbb{R}^d$ are the ground-truth regression weights for task $t$ which assign zero weights for the spurious latent factors. Note that

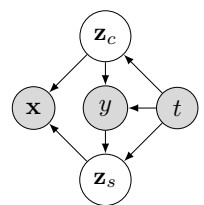

Figure 1: Assumed causal graph for the underlying data generating process.

there may be latent factors that are uncorrelated with $y$ in some tasks, which are treated as part of $\mathbf{z}_c^*$ but with zero regression weights. The spurious latent factors are assumed to be generated from the target variable with a different linear correlation function in each task $t$. Our goal is to recover the unobserved ground truth latent factors $\mathbf{z}^*$ given an empirical task distribution $p(t)$ over $N_t$ training tasks and an empirical data distribution $p(\mathbf{x}, y|t)$ for each task $t \in \{1, \cdots, N_t\}$.

Overall, our proposed method consists of two stages as illustrated in Figure 2. In the first stage, we train a multi-task neural network with a feature extractor shared across tasks and $N_t$ task-specific linear heads using maximum likelihood. We show that upon convergence, the representations learned by the feature extractor are identifiable up to some invertible linear transformation. In the second stage, we use the assumed causal structure across tasks to define a conditional prior over the underlying independent latent factors. We show that this enables simple maximum marginal likelihood learning for recovering the linear transformation in the representations obtained in the first stage, which reduces the identifiability class to permutations and scaling, and automatically disentangles and identifies the causes and effects of the target variable from the learned representations.

---

[1]Without loss of generality, we assume that $\mathbb{E}(y) = 0$. This can be achieved by standardizing $y$ in practice.

### 3.2 Stage 1: Multi-Task Regression Network (MTRN)

In the first stage, we train a multi-task regression network (MTRN) to recover the ground-truth latent factors up to some invertible linear transformation.

Let $f_{\boldsymbol{\phi},\mathbf{w}_t}(\mathbf{x}) = \mathbf{w}_t^\mathsf{T}\mathbf{h}_{\boldsymbol{\phi}}(\mathbf{x})$ be the output of an MTRN for task $t$, where $\mathbf{w}_t \in \mathbb{R}^d$ are the regression weights in the linear head for task $t$, and $\mathbf{h}_{\boldsymbol{\phi}}(\mathbf{x}) \in \mathbb{R}^d$ is the data representation produced by the feature extractor $\mathbf{h}_{\boldsymbol{\phi}}$ shared across all tasks with learnable parameters $\boldsymbol{\phi}$. As in typical non-linear regression settings, the likelihood is assumed to be Gaussian $p_{\boldsymbol{\theta}}(y|\mathbf{x}, t) = \mathcal{N}(y|f_{\boldsymbol{\phi},\mathbf{w}_t}(\mathbf{x}), \sigma_{r,t}^2)$ with mean modeled by an MTRN and variance fixed to some constant $\sigma_{r,t}^2$, where $\boldsymbol{\theta} := (\boldsymbol{\phi}, \mathbf{w}_1, \cdots, \mathbf{w}_{N_t})$ denotes all parameters in the MTRN. Following standard practice, we train the MTRN via maximum likelihood estimation (MLE):

$$\boldsymbol{\theta}' = \arg\max_{\boldsymbol{\theta}} \ \mathbb{E}_{p(t)p(\mathbf{x},y|t)}[\log p_{\boldsymbol{\theta}}(y|\mathbf{x}, t)]. \tag{1}$$

We first define linearly identifiable (or weakly identifiable) representations in the multi-task setting.

**Definition 3.1** (Multi-task weak identifiability). Let $\boldsymbol{\theta}$ and $\boldsymbol{\theta}'$ be two sets of parameters that satisfy (1). Then, the data representations are *linearly identifiable* if there exists an invertible matrix $\mathbf{A} \in \mathbb{R}^{d \times d}$ such that

$$p_{\boldsymbol{\theta}'}(y|\mathbf{x}, t) = p_{\boldsymbol{\theta}}(y|\mathbf{x}, t), \ \forall t, \mathbf{x}, y \quad \implies \quad \mathbf{h}_{\boldsymbol{\phi}'}(\mathbf{x}) = \mathbf{A}\mathbf{h}_{\boldsymbol{\phi}}(\mathbf{x}). \tag{2}$$

We show that data representations of MTRN are linearly identifiable if we have access to a set of sufficiently diverse tasks measured by the linear dependencies among their regression weights.

**Theorem 3.2.** *Let $\boldsymbol{\theta} := (\boldsymbol{\phi}, \mathbf{w}_1, \cdots, \mathbf{w}_{N_t})$ be a set of parameters that satisfy (1). Assume that $Span(Im(\mathbf{h}_{\boldsymbol{\phi}})) = \mathbb{R}^d$, i.e., the vectors in the image of $\mathbf{h}_{\boldsymbol{\phi}}$ span the whole $\mathbb{R}^d$. Suppose that there exist $d$ tasks $\{t_i\}_{i=1}^d \subseteq \{1, \cdots, N_t\}$ such that the set of regression weights $\{\mathbf{w}_{t_i}\}_{i=1}^d$ are linearly independent. Then, the data representations of MTRN are linearly identifiable.*

The proof of Theorem 3.2 can be found in Appendix A. Using Theorem 3.2, it is straightforward to show that MTRN trained with maximum likelihood estimation can recover the ground-truth data representations up to some invertible linear transformation.

**Corollary 3.3.** *Let $\mathbf{h}_* := \mathbf{f}_*^{-1} : \mathcal{X} \to \mathbb{R}^d$ be the inverse of the ground-truth mixing function $\mathbf{f}_*$, i.e., $\mathbf{z}^* = \mathbf{h}_*(\mathbf{x})$. Assume that $Span(Im(\mathbf{h}_*)) = \mathbb{R}^d$. Suppose that there exist $d$ tasks $\{t_i\}_{i=1}^d \subseteq \{1, \cdots, N_t\}$ such that the set of ground-truth regression weights $\{\mathbf{w}_{t_i}^*\}_{i=1}^d$ are linearly independent. Assume that (1) has a unique solution. Suppose that the optimization procedure for (1) converges to the optimal predictive likelihood under standard regularity conditions for consistency of MLE estimators, i.e.,*

$$p_{\boldsymbol{\theta}'}(y|\mathbf{x}, t) = p_*(y|\mathbf{x}, t) := \mathcal{N}(y|(\mathbf{w}_t^*)^T\mathbf{h}_*(\mathbf{x}), \sigma_{r,t}^2), \ \forall t, \mathbf{x}, y. \tag{3}$$

*Then, the feature extractor $\mathbf{h}_{\boldsymbol{\phi}'}$ is guaranteed to recover the ground truth data representations (or latent factors) up to some invertible linear transformation $\mathbf{A}_*$, i.e., $\mathbf{h}_{\boldsymbol{\phi}'}(\mathbf{x}) = \mathbf{A}_*\mathbf{h}_*(\mathbf{x})$.*

While Lachapelle et al. (2023)[Proposition 2.2] prove a similar proposition on MLE invariance to linear feature transformations, their proposition is built upon their Assumption 2.1 that the learned feature extractor $\mathbf{h}_{\boldsymbol{\phi}'}$ is linearly equivalent to the ground truth feature extractor $\mathbf{h}_*$. However, they do not specify under what conditions this assumption will hold for the MLE objective; they only specify conditions for the bi-level objective with a sparsity regularizer in their Section 3. In contrast, our Corollary 3.3 explicitly reveals such conditions for MLE, i.e., $Span(Im(\mathbf{h}_*)) = \mathbb{R}^d$ and the existence of $d$ linearly independent ground-truth task-specific vectors $\{\mathbf{w}_{t_i}^*\}_{i=1}^d$.

### 3.3 Stage 2: Multi-Task Linear Causal Model (MTLCM)

In the second stage, we fix the feature extractor $\mathbf{h}_{\boldsymbol{\phi}'}$ learned in the first stage and denote its representations by $\mathbf{h} := \mathbf{h}_{\boldsymbol{\phi}'}(\mathbf{x})$. Corollary 3.3 suggests that $\mathbf{h} = \mathbf{A}_*\mathbf{z}^*$ for some invertible matrix $\mathbf{A}_*$. We propose a *multi-task linear causal model* (MTLCM) to recover the ground-truth latent factors up permutations and scaling from $\mathbf{h}$ based on our assumed causal graph in Figure 1. The core idea of the MTLCM is to model the change in the causal and spurious latent factors across tasks with learnable task-specific parameters.

Let $\mathcal{T}(t) = \{\mathbf{c}_t, \boldsymbol{\gamma}_t\}$ be a collection of task-specific variables associated with task $t$, which are free parameters to be learned from data, where $\mathbf{c}_t \in \{0, 1\}^d$ are the causal indicator variables which determine the partition of $\mathbf{z} = \mathbf{z}_c \cup \mathbf{z}_s$ for the given task $t$ (i.e., $c_{t,i} = 1$ indicates that $z_i$ is a causal latent factor in task $t$ and $c_{t,i} = 0$ indicates that $z_i$ is a spurious latent factor in task $t$), and $\boldsymbol{\gamma}_t$ are the coefficients used to generate the spurious latent factors from $y$ for task $t$.

### 3.3.1 CONDITIONALLY FACTORIZED PRIOR GIVEN TASK AND TARGET VARIABLES

We assume that the causal latent factors $\mathbf{z}_c$ are sampled from a standard Gaussian distribution a priori:

$$p_{\mathcal{T}}(\mathbf{z}_c|t) = \mathcal{N}(\mathbf{z}_c|\mathbf{0}, \mathbf{I}), \tag{4}$$

which depends on the task variable $t$ since the causal indicator variable $\mathbf{c}_t$ that determines which subset of latent factors are causal varies across tasks.

According to the assumed data generating process, the target variable $y$ is a linear function of the data representations. Therefore, we assume that $y$ is generated from $\mathbf{z}_c$ via a linear Gaussian model with the regression weights $\mathbf{w}_t$ masked by the causal indicators $\mathbf{c}_t$:

$$p_{\mathcal{T}}(y|\mathbf{z}_c, t) = \mathcal{N}(y|(\mathbf{w}_t \circ \mathbf{c}_t)^{\mathrm{T}}\mathbf{z}, \sigma_p^2), \tag{5}$$

and that the spurious latent factors $\mathbf{z}_s$ are generated from $y$ via another linear Gaussian model:

$$p_{\mathcal{T}}(\mathbf{z}_s|y, t) = \mathcal{N}(\mathbf{z}_s|y\boldsymbol{\gamma}_t, \sigma_s^2\mathbf{I}). \tag{6}$$

The structured conditional prior over all latent factors given $t$ and $y$ that follows our assumed causal graph can be obtained by Bayes' Rule:

$$p_{\mathcal{T}}(\mathbf{z}|y, t) = \frac{p_{\mathcal{T}}(\mathbf{z}_c|t)p_{\mathcal{T}}(y|\mathbf{z}_c, t)p_{\mathcal{T}}(\mathbf{z}_s|y, t)}{\int p_{\mathcal{T}}(\mathbf{z}_c|t)p_{\mathcal{T}}(y|\mathbf{z}_c, t)p_{\mathcal{T}}(\mathbf{z}_s|y, t)d\mathbf{z}_s d\mathbf{z}_c}. \tag{7}$$

Since no prior knowledge of regression weights $\mathbf{w}_t$ is assumed, we marginalize out $\mathbf{w}_t$ from $p_{\mathcal{T}}(y|\mathbf{z}_c, t)$ under an uninformative prior (i.e., an infinite-variance Gaussian prior). This makes the structured conditional prior factorize over all latent factors (see Appendix D for a derivation):

$$p_{\mathcal{T}}(\mathbf{z}|y, t) = p_{\mathcal{T}}(\mathbf{z}_c|t)p_{\mathcal{T}}(\mathbf{z}_s|y, t) = \mathcal{N}(\mathbf{z}|\mathbf{a}_t, \boldsymbol{\Lambda}_t), \tag{8}$$

where the mean $\mathbf{a}_t$ and covariance $\boldsymbol{\Lambda}_t$ can be compactly expressed as:

$$\mathbf{a}_t := y\boldsymbol{\gamma}_t \circ (1 - \mathbf{c}_t) \quad \text{and} \quad \boldsymbol{\Lambda}_t := \mathrm{diag}(\sigma_s^2(1 - \mathbf{c}_t) + \mathbf{c}_t). \tag{9}$$

### 3.3.2 LINEAR GAUSSIAN LIKELIHOOD

Since the data representation $\mathbf{h}$ learned in the first stage is equivalent to $\mathbf{z}^*$ up to some linear transformation, we assume a linear Gaussian likelihood with invertible linear transformation $\mathbf{A}$:

$$p_{\mathbf{A}}(\mathbf{h}|\mathbf{z}) = \mathcal{N}(\mathbf{h}|\mathbf{Az}, \sigma_o^2\mathbf{I}), \tag{10}$$

where $\mathbf{A}$ is to be learned from data, which aims to recover the ground-truth linear transformation $\mathbf{A}_*$ for the linearly identifiable representation $\mathbf{h}$.

### 3.3.3 MAXIMUM MARGINAL LIKELIHOOD LEARNING

Let $\boldsymbol{\psi} = (\mathbf{A}, \mathcal{T})$ denote all parameters in an MTLCM, including the linear transformation $\mathbf{A}$ and the task-specific parameters $\mathcal{T}(t) = \{\mathbf{c}_t, \boldsymbol{\gamma}_t\}$ for all tasks $t$. The marginal likelihood for MTLCM is given by

$$p_{\boldsymbol{\psi}}(\mathbf{h}|y, t) = \int p_{\mathbf{A}}(\mathbf{h}|\mathbf{z})p_{\mathcal{T}}(\mathbf{z}|y, t)d\mathbf{z} = \mathcal{N}(\mathbf{h}|\boldsymbol{\mu}_t, \boldsymbol{\Sigma}_t), \tag{11}$$

where the mean $\boldsymbol{\mu}_t$ and covariance $\boldsymbol{\Sigma}_t$ have closed-form expressions:

$$\boldsymbol{\mu}_t = y\mathbf{A}(\boldsymbol{\gamma}_t \circ (1 - \mathbf{c}_t)) \quad \text{and} \quad \boldsymbol{\Sigma}_t = \mathbf{A}\mathrm{diag}(\sigma_s^2(1 - \mathbf{c}_t) + \mathbf{c}_t)\mathbf{A}^{\mathrm{T}} + \sigma_o^2\mathbf{I}. \tag{12}$$

It is important to note that in the single-task setting, the conditional prior $p(\mathbf{z}|y)$ over the latent factors $\mathbf{z}$ is non-factorized, since the causal latent factors $\mathbf{z}_c$ are parents of the target variable $y$, which become correlated when conditioning on $y$. In order to guarantee strong identifiability, iCaRL parameterizes

such non-factorized conditional priors using energy-based models optimized by variational inference and score matching, which turns out to be difficult to train in practice due to variational overpruning (Trippe & Turner, 2018) and high computational complexity (Hyvärinen & Dayan, 2005). In contrast, by conditioning on the task $t$ in addition to the target $y$ and leveraging the change in the causal/spurious latent factors across tasks, we obtain a conditionally factorized prior (8), which, together with the linear Gaussian likelihood (10), allows us to use maximum marginal likelihood learning to recover the ground-truth latent factors $\mathbf{z}^*$ up to permutations and scaling from the linearly identifiable data representations $\mathbf{h} = \mathbf{h}_\phi(\mathbf{x})$ learned in the first stage:

$$\psi' = \arg\max_{\psi} \ \mathbb{E}_{p(t)p(\mathbf{x},y|t)}[\log p_\psi(\mathbf{h}_\phi(\mathbf{x})|y,t)]. \tag{13}$$

It is worth noting that our method has greater applicability than the methods that rely on a learned probabilistic inverse $q_\psi(\mathbf{z}|\mathbf{x}, y)$ to extract identifiable latent factors from data such as iVAE and iCaRL, since $q_\psi(\mathbf{z}|\mathbf{x}, y)$ depends on the target variable $y$ which is often unknown at test time. In contrast, our method does not depend on $y$ at inference time, since the identifiable latent factors can be obtained by applying the inverse linear transformation learned by the MTLCM to the linearly identifiable data representations produced by the MTRN, i.e., $\mathbf{z} = \mathbf{A}^{-1}\mathbf{h}_\phi(\mathbf{x})$.

### 3.3.4 IDENTIFIABILITY THEORY

We first define the concept of strictly strong identifiability in the multi-task setting.

**Definition 3.4** (Strictly strong identifiability). Let $\psi$ and $\psi'$ be two sets of parameters that satisfy (13). The latent factors are identifiable up to permutations and scaling if there exists a permutation and scaling matrix $\mathbf{P} \in \mathbb{R}^{d \times d}$ such that

$$p_{\psi'}(\mathbf{h}|y,t) = p_\psi(\mathbf{h}|y,t), \ \forall \mathbf{h}, t, y, \quad \Longrightarrow \quad (\mathbf{A}')^{-1}\mathbf{h} = \mathbf{P}\mathbf{A}^{-1}\mathbf{h}. \tag{14}$$

We show that the latent factors of MTLCM are strictly strongly identifiable if there are sufficient variations of the causal/spurious latent factors across tasks measured by the linear dependencies among the natural parameters of their conditional priors.

**Theorem 3.5.** *Let $\mathbf{u} := [y, t]$ denote the conditioning variable and $k := 2d$. Assume that the learned and ground-truth linear transformations $\mathbf{A}$ and $\mathbf{A}_*$ are invertible. Suppose that there exist $k + 1$ points $\mathbf{u}_0, \mathbf{u}_1, \cdots, \mathbf{u}_k$ such that the matrix*

$$\mathbf{L} := [\boldsymbol{\eta}(\mathbf{u}_1) - \boldsymbol{\eta}(\mathbf{u}_0), \cdots, \boldsymbol{\eta}(\mathbf{u}_k) - \boldsymbol{\eta}(\mathbf{u}_0)] \tag{15}$$

*is invertible, where $\boldsymbol{\eta}(\mathbf{u}) := \begin{bmatrix} \boldsymbol{\Lambda}_t^{-1}\mathbf{a}_t \\ -\frac{1}{2}diag(\boldsymbol{\Lambda}_t) \end{bmatrix} \in \mathbb{R}^k$ are the natural parameters of $p_\mathcal{T}(\mathbf{z}|\mathbf{u})$. Assume that (13) has a unique solution. Suppose that the optimization procedure for (13) converges to the optimal marginal likelihood under standard regularity conditions for consistency of maximum marginal likelihood estimators, i.e.,*

$$p_{\psi'}(\mathbf{h}|y,t) = p_*(\mathbf{h}|y,t) := \mathcal{N}(\mathbf{h}|\boldsymbol{\mu}_t^*, \boldsymbol{\Sigma}_t^*), \quad \forall \mathbf{h}, y, t, \tag{16}$$

*where $\boldsymbol{\mu}_t^*$ and $\boldsymbol{\Sigma}_t^*$ are defined by Equation (12) but with the ground-truth linear transformation $\mathbf{A}_*$, ground-truth causal indicators $\mathbf{c}_t^*$ and ground-truth spurious coefficients $\boldsymbol{\gamma}_t^*$. Then, MTLCM is guaranteed to recover the ground-truth latent factors up to permutations and scaling.*

The proof of Theorem 3.5 can be found in Appendix B. The first part of the proof adapts the proof technique from Khemakhem et al. (2020a) to show identifiability up to block permutations and scaling. The second part of the proof is novel, which leverages the properties of the linear likelihood (10) to further reduce the block-identifiable equivalence class to permutations and scaling of the actual ground-truth latent factors.

## 4 EXPERIMENTS

This section empirically validates our model's ability to recover canonical representations up to permutations and scaling for both synthetic and real-world data. We contrast our model with the more general identifiable models of iVAE (Khemakhem et al., 2020a) and iCaRL (Lu et al., 2022). For a fair comparison, we also consider the multi-task extensions of iVAE and iCaRL, MT-iVAE and MT-iCaRL, which include the task variable $t$ in the conditioning variables $\mathbf{u}$ in their conditional priors $p_\mathcal{T}(\mathbf{z}|\mathbf{u})$, with the task-specific parameter $\mathcal{T}(t) = \{\mathbf{v}_t\}$ to be learned from data, which is the counterpart to $\mathcal{T}(t) = \{\mathbf{c}_t, \boldsymbol{\gamma}_t\}$ in our MTLCM but has no explicit interpretations with respect to a causal graph. Detailed model configurations can be found in Appendix C. Each experiment is run until convergence and repeated across 5 random seeds to guarantee reproducibility.

Table 1: Identifiability performance for recovering the linearly transformed synthetic latent factors measured by strong MCC (%).

| #Causal | 2 | | | | | 4 | | | |
|---|---|---|---|---|---|---|---|---|---|
| #Latent/Observed | 3/3 | 5/5 | 10/10 | 20/20 | 50/50 | 5/5 | 10/10 | 20/20 | 50/50 |
| iVAE | 87.75±5.02 | 78.02±0.73 | 81.36±0.57 | 82.30±0.27 | 81.96±0.07 | 81.67±2.97 | 74.29±0.30 | 77.57±0.15 | 79.79±0.10 |
| iCaRL | 75.22±6.40 | 74.55±2.09 | 72.37±2.22 | 79.43±0.52 | 80.00±1.00 | 66.98±1.32 | 66.00±3.00 | 71.54±1.69 | 78.67±0.61 |
| MT-iVAE | 91.78±8.12 | 90.14±5.01 | **99.89±0.04** | 97.90±1.51 | 90.56±3.18 | 76.09±7.69 | 76.36±2.32 | 98.42±0.88 | 94.53±2.49 |
| MT-iCaRL | 81.09±3.37 | 71.12±2.97 | 76.13±0.53 | 79.26±1.00 | 81.30±0.84 | 61.55±1.26 | 64.04±1.08 | 72.79±1.92 | 79.54±0.59 |
| MTLCM | **99.95±0.01** | **99.96±0.01** | 99.77±0.16 | **99.70±0.16** | **98.97±0.55** | **99.95±0.01** | **99.71±0.21** | **99.51±0.36** | **99.14±0.27** |

Table 2: Identifiability performance for recovering the non-linearly transformed synthetic latent factors measured by strong MCC (%). The weak MCC (%) for MTRN is also reported.

| #Causal | 4 | | | 8 | | | 12 | | |
|---|---|---|---|---|---|---|---|---|---|
| #Latent/Observed | 20/50 | 20/100 | 20/200 | 20/50 | 20/100 | 20/200 | 20/50 | 20/100 | 20/200 |
| iVAE | 73.11±1.13 | 77.42±0.20 | 76.95±0.31 | 65.18±1.49 | 68.66±0.14 | 69.05±0.17 | 58.70±0.33 | 60.33±0.27 | 59.85±0.31 |
| iCaRL | 56.70±3.49 | 63.29±4.26 | 58.64±2.83 | 57.09±2.41 | 60.66±2.74 | 61.02±2.43 | 52.93±2.13 | 58.80±1.81 | 54.40±2.54 |
| MT-iVAE | 71.78±1.45 | 80.14±0.37 | 73.89±2.98 | 65.44±1.60 | 69.31±0.35 | 68.56±0.34 | 55.79±1.61 | 60.56±0.23 | 59.61±0.30 |
| MT-iCaRL | 67.57±1.97 | 70.26±3.22 | 65.52±0.65 | 63.37±0.84 | 63.75±2.19 | 61.61±1.52 | 57.13±1.07 | 60.56±0.15 | 58.10±1.04 |
| MTLCM | **93.31±1.10** | **97.94±0.71** | **97.44±0.68** | **95.67±0.16** | **98.12±0.75** | **89.05±0.97** | **95.75±0.14** | **96.28±1.20** | **84.28±1.27** |
| MTRN (weak) | 89.38±0.71 | 96.15±0.91 | 96.19±0.87 | 93.96±0.22 | 97.63±0.79 | 87.75±0.99 | 95.14±0.17 | 96.12±1.27 | 83.70±1.22 |

## 4.1 SYNTHETIC DATA

We first validate our approach in the situation when the data generating process agrees with the assumptions of our models. For each task, we first sample the causal indicator variables $\mathbf{c}_t^*$. The causal latent factors $\mathbf{z}_c^*$ are then sampled from a standard Gaussian prior. These are then linearly combined according to random weights $\mathbf{w}_t^*$ to produce observed targets $y$ with a task-dependent noise corruption. Finally, the spurious variables $\mathbf{z}_s^*$ are generated via different weightings $\boldsymbol{\gamma}_t^*$ of the target $y$. This mirrors the causal data generating process described in Section 3. In Section 4.1.1, we generate observed data using random linear transformations of the ground-truth latent factors. In Section 4.1.2, we extend this to non-linear transformations parameterized by randomly initialized neural networks and demonstrate that our approach can be combined with the multi-task identifiability result up to linear transformations to recover permutations and scaling of the ground-truth. We also compare the learned causal indicator variables $\mathbf{c}_t$ with the ground-truth $\mathbf{c}_t^*$ and the causal discovery results from the conditional independence test (Chen, 2021; Lu et al., 2022) performed on the latent factors $\mathbf{z}$ recovered by our model. Detailed synthetic data generating process can be found in Appendix E.

### 4.1.1 LINEAR CASE

We study the ability of our proposed multi-task linear causal model (MTLCM) to recover the latent factors up to permutations and scaling via the Mean Correlation Coefficient (MCC) as in Khemakhem et al. (2020a). The synthetic data is generated by sampling 200 tasks of 100 samples each. Each task varies in its causal indicator variables $\mathbf{c}_t^*$, causal weights $\mathbf{w}_t^*$, and spurious coefficients $\boldsymbol{\gamma}_t^*$. We then transform the ground-truth latent factors $\mathbf{z}^*$ with a random invertible matrix $\mathbf{A}_*$ shared across all tasks to obtain linearly identifiable representations $\mathbf{h}$. Identifiability in this setting is assessed by directly computing the MCC score between the representations obtained from our MTLCM and the ground-truth latent factors, which is referred to as strong MCC. Since the data is known to be linearly identifiable, we use linear likelihoods for the baselines.

In Table 1, we show that MTLCM manages to recover the ground-truth latent factors from $\mathbf{h}$ up to permutations and scaling, and the result is scalable as the number of latent factors and the number of causal factors increase. In contrast, iVAE, iCaRL and their multi-task extensions underperform our model by a large margin in most cases. We also find that for all tasks, the learned causal indicator variables exactly match the ground-truth and the results from the conditional independence test. Ablation study for the effects of the learnable parameters and the type of the ground-truth transformation can be found in Appendix F.

### 4.1.2 NON-LINEAR CASE

A more general analysis of the identifiability of our proposed approach is to consider the extension of the linear experiments to the setting of *arbitrary* transformations of the latent factors. For this, we consider the case where random (non-linear) MLP neural networks are used to transform $\mathbf{z}^*$ into

Table 3: Identifiability performance for the latent factors learned on the superconductivity dataset measured by strong MCC (%). The weak MCC (%) for MTRN is also reported. "$-$" indicates divergence of optimization during training.

| Latent dim | 5 | 10 | 20 | 40 | 80 |
|---|---|---|---|---|---|
| iVAE | 32.87±1.16 | 33.21±1.04 | 30.68±0.39 | 37.41±0.84 | 45.52±0.81 |
| iCaRL | $-$ | 32.23±0.61 | 35.62±0.40 | 32.58±2.16 | 32.19±2.45 |
| MT-iVAE | 35.58±1.48 | 33.54±0.80 | 31.68±0.32 | 35.14±0.82 | 44.49±0.96 |
| MT-iCaRL | $-$ | $-$ | $-$ | $-$ | 42.26±2.33 |
| MTLCM | **98.90±0.03** | **96.93±0.12** | **84.56±1.11** | **46.31±0.34** | **48.94±2.16** |
| MTRN (weak) | 98.85±0.03 | 97.17±0.04 | 93.23±0.08 | 78.58±0.09 | 52.02±0.19 |

higher dimensional observations $\mathbf{x}$. By Corollary 3.3, it is possible to recover linearly identifiable representations $\mathbf{h}$ of the data by training standard multi-task regression networks (MTRNs). Identifiability in this setting is assessed by first performing a Canonical Correlation Analysis (CCA) as in Roeder et al. (2021), which linearly maps the obtained representations such that they maximize the covariance with the ground-truth latent factors. The resulting mapped representations can thus be compared with the ground-truth latent factors via the MCC score. This is referred to as weak MCC, which quantifies the linear identifiability of the learned representations from MTRNs. We further train our MTLCM on the linearly identifiable representations $\mathbf{h}$ obtained from the MTRN to obtain identifiable representations up to permutations and scaling. Identifiability in this setting is assessed by directly computing the MCC score between the representations obtained from our MTLCM and the ground-truth latent factors as in Section 4.1.1 (i.e., strong MCC). We assess this for various dimensionalities of the observed data and for different settings of the causal variables, where we generate 500 tasks of 200 samples each to improve convergence of the multitask model. The MTRN and the likelihoods in the baselines are parameterized by one-hidden-layer MLPs.

In Table 2, we find that the strong MCC for our MTLCM is able to match the weak MCC for the MTRN. In contrast, the strong MCC for iVAE, iCaRL and their multi-task extensions significantly underperform MTLCM. Again, we find that for all tasks, the learned causal indicator variables exactly match the ground-truth and the results from the conditional independence test.

## 4.2 REAL-WORLD DATA

We further evaluate our model on two real-world molecular datasets. We assume that the data $\mathbf{x}$ is generated by transforming some unknown ground-truth latent factors $\mathbf{z}^*$ with some unknown non-linear mixing function. Since $\mathbf{z}^*$ are unknown to us, identifiability in this setting is assessed by first training a model 5 times with different random seeds and initializations then computing the MCC score between the data representations recovered by each pair of those 5 models, as in Khemakhem et al. (2020b). As in Section 4.1, we employ the weak MCC score to assess the linear identifiability of the representations $\mathbf{h}$ learned by the MTRN and the strong MCC score to assess the strictly strong identifiability of the latent factors $\mathbf{z}$ recovered by our MTLCM and the baselines.

### 4.2.1 SUPERCONDUCTIVITY

The superconductivity dataset (Hamidieh, 2018) consists of $21,263$ superconductors. We consider the tasks of regressing $80$ readily computed target features such as mean atomic mass, thermal conductivity and valence of the superconductors from their chemical formulae, represented as discrete counts of the atoms present in the molecule. The MTRN and the likelihoods in the baselines are parameterized by MLP neural networks.

In Table 3, we find that the strong MCC for our MTLCM is greater than $0.96$ and is able to match the weak MCC for the MTRN when the dimensions of the latent representations are 5 and 10, showing that our method manages to recover canonical latent representations for the superconductors. Interestingly, the strong MCC score for the MTLCM decreases as we increase the number of latent factors in the model, suggesting that there are at most 10-20 independent tasks out of the 80 targets used for this data. In sharp contrast, all baseline models fail to recover identifiable latent factors for the superconductors in all cases as their strong MCC scores do not exceed $0.4$. In addition, there are several settings where optimization diverged during training, since VAE-based models are generally difficult to train on discrete inputs of chemical formulae.

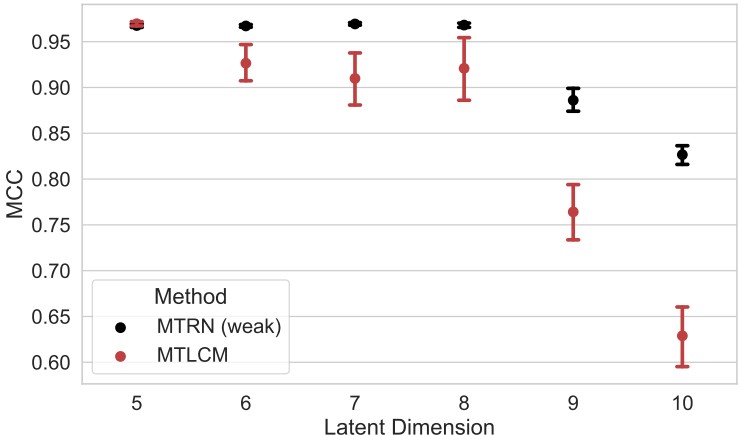

Figure 3: Identifiability performance for the latent factors learned on the QM9 dataset.

### 4.2.2 QM9

The QM9 dataset (Ruddigkeit et al., 2012; Ramakrishnan et al., 2014) is a commonly used benchmark for molecular prediction tasks consisting of $134,000$ enumerated organic molecules of up to nine heavy atoms together with a set of 12 calculated quantum chemical properties. In contrast to the more artificial superconductivity dataset considered in Section 4.2.1, the QM9 dataset enables us to assess the feasibility of achieving identifiable representations in the context of highly non-trivial quantum chemical properties which are highly relevant to their pharmacological profile. Accurately modeling this dataset requires us to capture potential three-dimensional atomic interactions, allowing us to assess the translation of our theoretical results to more complex equivariant graph neural network architectures. For this reason, we use an equivariant graph neural network (EGNN) (Satorras et al., 2021) as the feature extractor for the MTRN. This enables the model to incorporate positional features of each atom while exhibiting equivariance to their rotation, translation or reflection. Given that the graph autoencoders proposed in Satorras et al. (2021) and prior works (Kipf & Welling, 2016; Simonovsky & Komodakis, 2018; Liu et al., 2019) do not provide a means of jointly decoding the feature and adjacency matrices, we do not consider the iVAE and iCaRL baselines for this dataset.

In Figure 3, the weak identifiability achieved from the MTRN implies that identifiability is achievable up to eight latent features, suggesting there may be some redundancies between tasks, after which there is a gradual decline in MCC. Nonetheless, the MTLCM is able to closely approximate the weak MCC score up to eight latent factors, always surpassing a score of 0.9, demonstrating its ability to recover permutation identifiable representations in the context of realistic molecular datasets.

## 5 CONCLUSION

We have proposed a novel perspective on the problem of identifiable representations by exploring the implications of explicitly modeling task structures. We have shown that this implies new identifiability results, in particular for linear equivalence classes in the general case of multi-task regression. Furthermore, while spurious correlations have been shown to be a failure case of deep learning in many recent works, we have demonstrated that such latent spurious signals may in fact be leveraged to *improve* the ability of a model to recover more robust disentangled representations. In particular, we have shown that when the latent space is explicitly represented as consisting of a partitioning of causal and spurious features per task, the linear identifiability result of the multi-task setting may be reduced to identifiability up to simple permutations and scaling. Finally, we have confirmed that the theoretical results hold both for the synthetic data where our model's assumptions are satisfied and for real-world molecular datasets of superconductors and organic small molecules. We anticipate that this may reveal new research directions for the study of both causal representations and synergies with multi-task methods.

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

## A    PROOF OF THEOREM 3.2

*Proof.* By the assumption that the predictive likelihoods for the two sets of parameters $\boldsymbol{\theta}'$ and $\boldsymbol{\theta}$ are equal, we have

$$p_{\boldsymbol{\theta}'}(y|\mathbf{x}, t) = p_{\boldsymbol{\theta}}(y|\mathbf{x}, t), \quad \forall t, \mathbf{x}, y, \tag{17}$$

$$\implies \mathcal{N}(y|f_{\boldsymbol{\phi}', \mathbf{w}_t'}(\mathbf{x}), \sigma_{r,t}^2) = \mathcal{N}(y|f_{\boldsymbol{\phi}, \mathbf{w}_t}(\mathbf{x}), \sigma_{r,t}^2), \quad \forall t, \mathbf{x}, y, \tag{18}$$

$$\implies \mathcal{N}(y|\mathbf{h}_{\boldsymbol{\phi}'}(\mathbf{x})^\mathsf{T}\mathbf{w}_t', \sigma_{r,t}^2) = \mathcal{N}(y|\mathbf{h}_{\boldsymbol{\phi}}(\mathbf{x})^\mathsf{T}\mathbf{w}_t, \sigma_{r,t}^2), \quad \forall t, \mathbf{x}, y. \tag{19}$$

This implies that the means of the two Gaussian likelihoods on both sides are identical:

$$\mathbf{h}_{\boldsymbol{\phi}'}(\mathbf{x})^\mathsf{T}\mathbf{w}_t' = \mathbf{h}_{\boldsymbol{\phi}}(\mathbf{x})^\mathsf{T}\mathbf{w}_t, \quad \forall t, \mathbf{x}, y. \tag{20}$$

By the assumption that $\mathrm{Span}(\mathrm{Im}(\mathbf{h}_{\boldsymbol{\phi}})) = \mathbb{R}^d$, there exist $d$ inputs $\mathbf{x}_1, \cdots, \mathbf{x}_d$ such that the matrix $\mathbf{H} = [\mathbf{h}_{\boldsymbol{\phi}}(\mathbf{x}_1), \cdots, \mathbf{h}_{\boldsymbol{\phi}}(\mathbf{x}_d)] \in \mathbb{R}^{d \times d}$ is invertible. By the assumption that there exist $d$ tasks $\{t_i\}_{i=1}^d$ such that the set of regression weights $\{\mathbf{w}_{t_i}\}_{i=1}^d$ are linearly independent, we construct an invertible matrix $\mathbf{W} = [\mathbf{w}_{t_1}, \cdots, \mathbf{w}_{t_d}] \in \mathbb{R}^{d \times d}$. For $\mathbf{h}_{\boldsymbol{\phi}'}$, we similarly define $\mathbf{H}' = [\mathbf{h}_{\boldsymbol{\phi}'}(\mathbf{x}_1), \cdots, \mathbf{h}_{\boldsymbol{\phi}'}(\mathbf{x}_d)] \in \mathbb{R}^{d \times d}$ and $\mathbf{W}' = [\mathbf{w}_{t_1}', \cdots, \mathbf{w}_{t_d}'] \in \mathbb{R}^{d \times d}$.

Now, we evaluate Equation (20) at the $d$ inputs $\mathbf{x}_1, \cdots, \mathbf{x}_d$ and $d$ tasks $t_1, \cdots, t_d$ defined above, which gives us the following linear equation:

$$(\mathbf{H}')^\mathsf{T}\mathbf{W}' = \mathbf{H}\mathbf{W}. \tag{21}$$

Since both $\mathbf{H}$ and $\mathbf{W}$ are invertible by assumption and the weight matrices $\mathbf{W}$ and $\mathbf{W}'$ do not depend on the input $\mathbf{x}$, the matrix $\mathbf{W}'$ must be invertible.

Now, evaluating Equation (20) at the $d$ tasks $t_1, \cdots, t_d$, we have

$$(\mathbf{W}')^\mathsf{T}\mathbf{h}_{\boldsymbol{\phi}'}(\mathbf{x}) = \mathbf{W}^\mathsf{T}\mathbf{h}_{\boldsymbol{\phi}}(\mathbf{x}), \quad \forall \mathbf{x} \tag{22}$$

$$\implies \mathbf{h}_{\boldsymbol{\phi}'}(\mathbf{x}) = (\mathbf{W}')^{-\mathsf{T}}\mathbf{W}^\mathsf{T}\mathbf{h}_{\boldsymbol{\phi}}(\mathbf{x}), \quad \forall \mathbf{x} \tag{23}$$

$$\implies \mathbf{h}_{\boldsymbol{\phi}'}(\mathbf{x}) = \mathbf{A}\mathbf{h}_{\boldsymbol{\phi}}(\mathbf{x}), \quad \forall \mathbf{x}. \tag{24}$$

Note that we have shown that $\mathbf{A} := (\mathbf{W}')^{-\mathsf{T}}\mathbf{W}^\mathsf{T}$ is invertible. This completes the proof. $\quad\square$

## B    PROOF OF THEOREM 3.5

*Proof.* Let $k := 2d$ and $\mathbf{u} := [y, t]$. We first rewrite the density of the conditional prior in the exponential family form:

$$p_{\mathcal{T}}(\mathbf{z}|\mathbf{u}) = Z(\mathbf{u})^{-1} \exp\left(\mathbf{T}(\mathbf{z})^\mathsf{T}\boldsymbol{\eta}(\mathbf{u})\right), \tag{25}$$

where $Z(\mathbf{u}) = (2\pi)^{d/2}|\boldsymbol{\Lambda}_t|^{0.5} \exp\left(-\frac{1}{2}\mathbf{a}_t^\mathsf{T}\boldsymbol{\Lambda}_t\mathbf{a}_t\right)$ is the normalizing constant, $\mathbf{T}(\mathbf{z}) = \begin{bmatrix} \mathbf{z} \\ \mathbf{z} \circ \mathbf{z} \end{bmatrix} \in \mathbb{R}^k$ are the sufficient statistics, and $\boldsymbol{\eta}(\mathbf{u}) = \begin{bmatrix} \boldsymbol{\Lambda}_t^{-1}\mathbf{a}_t \\ -\frac{1}{2}\mathrm{diag}(\boldsymbol{\Lambda}_t) \end{bmatrix} \in \mathbb{R}^k$ are the natural parameters. We also rewrite the likelihood $p_\mathbf{A}(\mathbf{h}|\mathbf{z})$ using the noise distribution $p_{\boldsymbol{\epsilon}_o}(\boldsymbol{\epsilon}_o) = \mathcal{N}(\boldsymbol{\epsilon}_o|\mathbf{0}, \sigma_o^2\mathbf{I})$:

$$p_\mathbf{A}(\mathbf{h}|\mathbf{z}) = \mathcal{N}(\mathbf{h}|\mathbf{A}\mathbf{z}, \sigma_o^2\mathbf{I}) = \mathcal{N}(\mathbf{h} - \mathbf{A}\mathbf{z}|\mathbf{0}, \sigma_o^2\mathbf{I}) = p_{\boldsymbol{\epsilon}_o}(\mathbf{h} - \mathbf{A}\mathbf{z}). \tag{26}$$

Let $\mathbf{A}_*$ be the ground-truth transformation matrix such that $\mathbf{z}^* = \mathbf{A}_*^{-1}\mathbf{h}$, and $\mathcal{T}_*(t) = \{\mathbf{c}_t^*, \boldsymbol{\gamma}_t^*\}$ the ground-truth task-specific variables associated with each task $t$. The proof starts off by using the fact that we have maximized the marginal likelihood (11) of $\mathbf{A}$ and $\mathcal{T}$ for all tasks. This means that the marginal likelihoods of the two models are identical:

$$p_{\mathbf{A}, \mathcal{T}}(\mathbf{h}|\mathbf{u}) = p_{\mathbf{A}_*, \mathcal{T}_*}(\mathbf{h}|\mathbf{u}), \quad \forall \mathbf{h}, \mathbf{u}. \tag{27}$$

The goal is to show that the latent factors $\mathbf{z} = \mathbf{A}^{-1}\mathbf{h}$ recovered by our model and the ground-truth latent factor $\mathbf{z}^* = \mathbf{A}_*^{-1}\mathbf{h}$ are identical up to permutations and scaling for all $\mathbf{h}$.

Starting from the equality of the two marginal likelihoods (27), we have

$$p_{\mathbf{A},\mathcal{T}}(\mathbf{h}|\mathbf{u}) = p_{\mathbf{A}_*,\mathcal{T}_*}(\mathbf{h}|\mathbf{u}) \tag{28}$$

$$\iff \int p_{\mathbf{A}}(\mathbf{h}|\mathbf{z})p_{\mathcal{T}}(\mathbf{z}|\mathbf{u})d\mathbf{z} = \int p_{\mathbf{A}_*}(\mathbf{h}|\mathbf{z})p_{\mathcal{T}_*}(\mathbf{z}|\mathbf{u})d\mathbf{z} \tag{29}$$

$$\iff \int p_{\boldsymbol{\epsilon}_o}(\mathbf{h} - \mathbf{A}\mathbf{z})p_{\mathcal{T}}(\mathbf{z}|\mathbf{u})d\mathbf{z} = \int p_{\boldsymbol{\epsilon}_o}(\mathbf{h} - \mathbf{A}_*\mathbf{z})p_{\mathcal{T}_*}(\mathbf{z}|\mathbf{u})d\mathbf{z} \tag{30}$$

$$\iff \int p_{\boldsymbol{\epsilon}_o}(\mathbf{h} - \bar{\mathbf{h}})p_{\mathcal{T}}(\mathbf{A}^{-1}\bar{\mathbf{h}}|\mathbf{u})\det(\mathbf{A})^{-1}d\bar{\mathbf{h}} = \int p_{\boldsymbol{\epsilon}_o}(\mathbf{h} - \hat{\mathbf{h}})p_{\mathcal{T}_*}(\mathbf{A}_*^{-1}\hat{\mathbf{h}}|\mathbf{u})\det(\mathbf{A}_*)^{-1}d\hat{\mathbf{h}} \tag{31}$$

$$\iff \int p_{\boldsymbol{\epsilon}_o}(\mathbf{h} - \bar{\mathbf{h}})\tilde{p}_{\mathbf{A},\mathcal{T},\mathbf{u}}(\bar{\mathbf{h}})d\bar{\mathbf{h}} = \int p_{\boldsymbol{\epsilon}_o}(\mathbf{h} - \hat{\mathbf{h}})\tilde{p}_{\mathbf{A}_*,\mathcal{T}_*,\mathbf{u}}(\hat{\mathbf{h}})d\hat{\mathbf{h}} \tag{32}$$

$$\iff (p_{\boldsymbol{\epsilon}_o} * \tilde{p}_{\mathbf{A},\mathcal{T},\mathbf{u}})(\mathbf{h}) = (p_{\boldsymbol{\epsilon}_o} * \tilde{p}_{\mathbf{A}_*,\mathcal{T}_*,\mathbf{u}})(\mathbf{h}) \tag{33}$$

$$\iff F[p_{\boldsymbol{\epsilon}_o}](\boldsymbol{\omega})F[\tilde{p}_{\mathbf{A},\mathcal{T},\mathbf{u}}](\boldsymbol{\omega}) = F[p_{\boldsymbol{\epsilon}_o}](\boldsymbol{\omega})F[\tilde{p}_{\mathbf{A}_*,\mathcal{T}_*,\mathbf{u}}](\boldsymbol{\omega}) \tag{34}$$

$$\iff F[\tilde{p}_{\mathbf{A},\mathcal{T},\mathbf{u}}](\boldsymbol{\omega}) = F[\tilde{p}_{\mathbf{A}_*,\mathcal{T}_*,\mathbf{u}}](\boldsymbol{\omega}) \tag{35}$$

$$\iff \tilde{p}_{\mathbf{A},\mathcal{T},\mathbf{u}}(\mathbf{h}) = \tilde{p}_{\mathbf{A}_*,\mathcal{T}_*,\mathbf{u}}(\mathbf{h}) \tag{36}$$

$$\iff p_{\mathcal{T}}(\mathbf{A}^{-1}\mathbf{h}|\mathbf{u})\det(\mathbf{A})^{-1} = p_{\mathcal{T}_*}(\mathbf{A}_*^{-1}\mathbf{h}|\mathbf{u})\det(\mathbf{A}_*)^{-1} \tag{37}$$

$$\iff \mathbf{T}(\mathbf{A}^{-1}\mathbf{h})^{\mathsf{T}}\boldsymbol{\eta}(\mathbf{u}) - \log Z(\mathbf{u}) - \log\det(\mathbf{A}) = \mathbf{T}(\mathbf{A}_*^{-1}\mathbf{h})^{\mathsf{T}}\boldsymbol{\eta}_*(\mathbf{u}) - \log Z_*(\mathbf{u}) - \log\det(\mathbf{A}_*), \tag{38}$$

where

- Equation (31) follows by the definition $\bar{\mathbf{h}} := \mathbf{A}\mathbf{z}, \hat{\mathbf{h}} := \mathbf{A}_*\mathbf{z}$,

- Equation (32) follows by the definition $\tilde{p}_{\mathbf{A},\mathcal{T},\mathbf{u}}(\bar{\mathbf{h}}) := p_{\mathcal{T}}(\mathbf{A}^{-1}\bar{\mathbf{h}}|\mathbf{u})\det(\mathbf{A})^{-1}$,

- $*$ in Equation (33) denotes the convolution operator,

- $F$ in Equation (34) denotes the Fourier transform operator,

- Equation (35) follows since the characteristic function $F[p_{\boldsymbol{\epsilon}_o}]$ of the Gaussian noise $\boldsymbol{\epsilon}_o$ is nonzero almost everywhere.

Now we evaluate Equation 38 at $\mathbf{u} = \mathbf{u}_0, \mathbf{u}_1, \cdots, \mathbf{u}_k$ from our assumption to obtain $k + 1$ such equations and subtract the first equation from the remaining $k$ equations to obtain the following $k$ equations:

$$\mathbf{T}(\mathbf{A}^{-1}\mathbf{h})^{\mathsf{T}}(\boldsymbol{\eta}(\mathbf{u}_l) - \boldsymbol{\eta}(\mathbf{u}_0)) + \log\frac{Z(\mathbf{u}_0)}{Z(\mathbf{u}_l)} = \mathbf{T}(\mathbf{A}_*^{-1}\mathbf{h})^{\mathsf{T}}(\boldsymbol{\eta}_*(\mathbf{u}_l) - \boldsymbol{\eta}_*(\mathbf{u}_0)) + \log\frac{Z_*(\mathbf{u}_0)}{Z_*(\mathbf{u}_l)}, \tag{39}$$

where $l = 1, \cdots, k$. Putting those $k$ equations in the matrix-vector form gives

$$\mathbf{L}^{\mathsf{T}}\mathbf{T}(\mathbf{A}^{-1}\mathbf{h}) = \mathbf{L}_*^{\mathsf{T}}\mathbf{T}(\mathbf{A}_*^{-1}\mathbf{h}) + \mathbf{q}, \tag{40}$$

where $q_l = \log\frac{Z_*(\mathbf{u}_0)Z(\mathbf{u}_l)}{Z_*(\mathbf{u}_l)Z(\mathbf{u}_0)}$, $\mathbf{L}$ is the invertible matrix defined in the assumption, and $\mathbf{L}_*$ is similarly defined for the second model. Since $\mathbf{L}$ is invertible, we can left multiply Equation (40) by $\mathbf{L}^{-\mathsf{T}}$ to obtain

$$\mathbf{T}(\mathbf{A}^{-1}\mathbf{h}) = \mathbf{M}\mathbf{T}(\mathbf{A}_*^{-1}\mathbf{h}) + \mathbf{r}, \tag{41}$$

where $\mathbf{M} = \mathbf{L}^{-\mathsf{T}}\mathbf{L}_*^{\mathsf{T}}$ and $\mathbf{r} = \mathbf{L}^{-\mathsf{T}}\mathbf{q}$. We note that our assumption only says $\mathbf{L}$ is invertible and tells us nothing about $\mathbf{L}_*$. Therefore, we need to show that $\mathbf{M}$ is invertible. Let $\mathbf{h}_l := \mathbf{A}\mathbf{z}_l, l = 0, \cdots, k$. We evaluate Equation (41) at these $k + 1$ points to obtain $k + 1$ such equations, and subtract the first equation from the remaining $k$ equations. This gives us

$$[\mathbf{T}(\mathbf{z}_1) - \mathbf{T}(\mathbf{z}_0), \cdots, \mathbf{T}(\mathbf{z}_k) - \mathbf{T}(\mathbf{z}_0)] = \mathbf{M}[\mathbf{T}(\mathbf{A}_*^{-1}\mathbf{h}_1) - \mathbf{T}(\mathbf{A}_*^{-1}\mathbf{h}_0), \cdots, \mathbf{T}(\mathbf{A}_*^{-1}\mathbf{h}_k) - \mathbf{T}(\mathbf{A}_*^{-1}\mathbf{h}_0)]. \tag{42}$$

We denote Equation (42) by $\mathbf{R} := \mathbf{M}\mathbf{R}_*$. We need to show that for any given $\mathbf{z}_0$, there exist $k$ points $\mathbf{z}_1, \cdots, \mathbf{z}_k$ such that the columns of $\mathbf{R}$ are linearly independent. Suppose, for contradiction,

that the columns of $\mathbf{R}$ would never be linearly independent for any $\mathbf{z}_1, \cdots, \mathbf{z}_k$. Then the function $\mathbf{g}(\mathbf{z}) := \mathbf{T}(\mathbf{z}) - \mathbf{T}(\mathbf{z}_0)$ would live in a $k-1$ or lower dimensional subspace, and therefore we would be able to find a non-zero vector $\boldsymbol{\lambda} \in \mathbb{R}^k$ orthogonal to that subspace. This would imply that $(\mathbf{T}(\mathbf{z}) - \mathbf{T}(\mathbf{z}_0))^\mathsf{T} \boldsymbol{\lambda} = \mathbf{0}$ and thus $\mathbf{T}(\mathbf{z})^\mathsf{T} \boldsymbol{\lambda} = \mathbf{T}(\mathbf{z}_0)^\mathsf{T} \boldsymbol{\lambda} = const, \ \forall \mathbf{z}$, which contradicts the fact that our conditionally factorized multivariate Gaussian prior $p_{\mathcal{T}}(\mathbf{z}|\mathbf{u})$ is strongly exponential (see Khemakhem et al. (2020a) for the definition). This shows that there exist $k$ points $\mathbf{z}_1, \cdots, \mathbf{z}_k$ such that the columns of $\mathbf{R}$ are linearly independent for any given $\mathbf{z}_0$. Therefore, $\mathbf{R}$ is invertible. Since $\mathbf{R} = \mathbf{M}\mathbf{R}_*$ and $\mathbf{M}$ is not a function of $\mathbf{z}$, this tells us that $\mathbf{M}$ must be invertible.

Now that we have shown that $\mathbf{M}$ is invertible, the next step is to show that $\mathbf{M}$ is a block transformation matrix. We define a linear function $\mathbf{l}(\mathbf{z}) = \mathbf{A}_*^{-1}\mathbf{A}\mathbf{z}$. Now, Equation (41) becomes

$$\mathbf{T}(\mathbf{z}) = \mathbf{M}\mathbf{T}(\mathbf{l}(\mathbf{z})) + \mathbf{r}. \tag{43}$$

We first show that the linear function $\mathbf{l}$ is a point-wise function. We differentiate both sides of the above equation w.r.t. $z_s$ and $z_t$ ($\forall s \neq t$) to obtain:

$$\frac{\partial \mathbf{T}(\mathbf{z})}{\partial z_s} = \mathbf{M}\sum_{i=1}^{d} \frac{\partial \mathbf{T}(\mathbf{l}(\mathbf{z}))}{\partial l_i(\mathbf{z})} \frac{\partial l_i(\mathbf{z})}{\partial z_s}, \tag{44}$$

$$\frac{\partial^2 \mathbf{T}(\mathbf{z})}{\partial z_s \partial z_t} = \mathbf{M}\sum_{i=1}^{d}\sum_{j=1}^{d} \frac{\partial^2 \mathbf{T}(\mathbf{l}(\mathbf{z}))}{\partial l_i(\mathbf{z})\partial l_j(\mathbf{z})} \frac{\partial l_j(\mathbf{z})}{\partial z_t} \frac{\partial l_i(\mathbf{z})}{\partial z_s} + \mathbf{M}\sum_{i=1}^{d} \frac{\partial \mathbf{T}(\mathbf{l}(\mathbf{z}))}{\partial l_i(\mathbf{z})} \frac{\partial^2 l_i(\mathbf{z})}{\partial z_s \partial z_t}. \tag{45}$$

Since the prior $p_{\mathcal{T}}(\mathbf{z}|\mathbf{u})$ is conditionally factorized, the second-order cross derivatives of the sufficient statistics are zeros. Therefore, the second equation above can be simplified as follows:

$$\mathbf{0} = \frac{\partial^2 \mathbf{T}(\mathbf{z})}{\partial z_s \partial z_t} \tag{46}$$

$$= \mathbf{M}\sum_{i=1}^{d} \frac{\partial^2 \mathbf{T}(\mathbf{l}(\mathbf{z}))}{\partial l_i(\mathbf{z})^2} \frac{\partial l_i(\mathbf{z})}{\partial z_t} \frac{\partial l_i(\mathbf{z})}{\partial z_s} + \mathbf{M}\sum_{i=1}^{d} \frac{\partial \mathbf{T}(\mathbf{l}(\mathbf{z}))}{\partial l_i(\mathbf{z})} \frac{\partial^2 l_i(\mathbf{z})}{\partial z_s \partial z_t} \tag{47}$$

$$= \mathbf{M}\mathbf{T}''(\mathbf{z})\mathbf{l}'_{s,z}(\mathbf{z}) + \mathbf{M}\mathbf{T}'(\mathbf{z})\mathbf{l}''_{s,z}(\mathbf{z}) \tag{48}$$

$$= \mathbf{M}\mathbf{T}'''(\mathbf{z})\mathbf{l}'''_{s,z}(\mathbf{z}), \tag{49}$$

where

$$\mathbf{T}''(\mathbf{z}) = \left[\frac{\partial^2 \mathbf{T}(\mathbf{l}(\mathbf{z}))}{\partial l_1(\mathbf{z})^2}, \cdots, \frac{\partial^2 \mathbf{T}(\mathbf{l}(\mathbf{z}))}{\partial l_d(\mathbf{z})^2}\right] \in \mathbb{R}^{k \times d}, \tag{50}$$

$$\mathbf{l}'_{s,z}(\mathbf{z}) = \left[\frac{\partial l_1(\mathbf{z})}{\partial z_t} \frac{\partial l_1(\mathbf{z})}{\partial z_s}, \cdots, \frac{\partial l_d(\mathbf{z})}{\partial z_t} \frac{\partial l_d(\mathbf{z})}{\partial z_s}\right]^\mathsf{T} \in \mathbb{R}^d, \tag{51}$$

$$\mathbf{T}'(\mathbf{z}) = \left[\frac{\partial \mathbf{T}(\mathbf{l}(\mathbf{z}))}{\partial l_1(\mathbf{z})}, \cdots, \frac{\partial \mathbf{T}(\mathbf{l}(\mathbf{z}))}{\partial l_d(\mathbf{z})}\right] \in \mathbb{R}^{k \times d}, \tag{52}$$

$$\mathbf{l}''_{s,z}(\mathbf{z}) = \left[\frac{\partial^2 l_1(\mathbf{z})}{\partial z_s \partial z_t}, \cdots, \frac{\partial^2 l_d(\mathbf{z})}{\partial z_s \partial z_t}\right]^\mathsf{T} \in \mathbb{R}^d, \tag{53}$$

$$\mathbf{T}'''(\mathbf{z}) = [\mathbf{T}''(\mathbf{z}), \mathbf{T}'(\mathbf{z})] \in \mathbb{R}^{k \times k}, \tag{54}$$

$$\mathbf{l}'''_{s,z}(\mathbf{z}) = [\mathbf{l}'_{s,z}(\mathbf{z})^\mathsf{T}, \mathbf{l}''_{s,z}(\mathbf{z})^\mathsf{T}]^\mathsf{T} \in \mathbb{R}^k. \tag{55}$$

By Lemma 5 in Khemakhem et al. (2020a) and the fact that $k = 2d$, we have that the rank of $\mathbf{T}'''(\mathbf{z})$ is $2d$ and thus it is invertible for all $\mathbf{z}$. Since $\mathbf{M}$ is also invertible, we have that $\mathbf{M}\mathbf{T}'''(\mathbf{z})$ is invertible. Since $\mathbf{M}\mathbf{T}'''(\mathbf{z})\mathbf{l}'''_{s,z}(\mathbf{z}) = \mathbf{0}$, it must be that $\mathbf{l}'''_{s,z}(\mathbf{z}) = \mathbf{0}, \ \forall \mathbf{z}$. In particular, this means that $\mathbf{l}'_{s,z}(\mathbf{z}) = \mathbf{0}, \ \forall s \neq t$ for all $\mathbf{z}$, which shows that the linear function $\mathbf{l}(\mathbf{z}) = \mathbf{A}_*^{-1}\mathbf{A}\mathbf{z}$ is a point-wise linear function.

Now, we are ready to show that $\mathbf{M}$ is a block transformation matrix. Without loss of generality, we assume that the permutation in the point-wise linear function $\mathbf{l}$ is the identity. That is, $\mathbf{l}(\mathbf{z}) = [l_1 z_1, \cdots, l_d z_d]^\mathsf{T}$ for some linear univariate scalars $l_1, \cdots, l_d \in \mathbb{R}$. Since $\mathbf{A}$ and $\mathbf{A}_*$ are invertible, we have that $\mathbf{l}^{-1}(\mathbf{z}) = [l_1^{-1} z_1, \cdots, l_d^{-1} z_d]^\mathsf{T}$. Define

$$\bar{\mathbf{T}}(\mathbf{l}(\mathbf{z})) := \mathbf{T}(\mathbf{l}(\mathbf{z})) + \mathbf{M}^{-1}\mathbf{r} \tag{56}$$

and plug it into Equation (43) gives:

$$\mathbf{T}(\mathbf{z}) = \mathbf{M}\bar{\mathbf{T}}(\mathbf{l}(\mathbf{z})). \tag{57}$$

We then apply $\mathbf{l}^{-1}$ to $\mathbf{z}$ at both sides of the Equation (57) to obtain

$$\mathbf{T}(\mathbf{l}^{-1}(\mathbf{z})) = \mathbf{M}\bar{\mathbf{T}}(\mathbf{z}). \tag{58}$$

Since $\mathbf{l}$ is a point-wise function, for a given $q \in \{1, \cdots, k\}$ we have that

$$0 = \frac{\partial \mathbf{T}(\mathbf{l}^{-1}(\mathbf{z}))_q}{\partial z_s} = \sum_{j=1}^{k} M_{q,j} \frac{\partial \bar{\mathbf{T}}(\mathbf{z})_j}{\partial z_s}, \quad \text{for any } s \text{ such that } q \neq s \text{ and } q \neq 2s. \tag{59}$$

Since the entries in $\bar{\mathbf{T}}(\mathbf{z})$ are linearly independent, it must be that $M_{q,j} = 0$ for any $j$ such that $\frac{\partial \bar{\mathbf{T}}(\mathbf{z})_j}{\partial z_s} \neq 0$. This includes the entries $j$ in $\bar{\mathbf{T}}(\mathbf{z})$ which depend on $z_s$ (i.e., $j = s$ and $j = 2s$). Note that this holds true for any $s$ such that $q \neq s$ and $q \neq 2s$. Therefore, when $q$ is the index of an entry in the sufficient statistics $\mathbf{T}$ that corresponds to $z_i$ (i.e., $q = i$ or $q = 2i$, and $i \neq s$), the only possible non-zero $M_{q,j}$ for $j$ are the ones that map between $\mathbf{T}_i(z_i)$ and $\bar{\mathbf{T}}_i(l_i(z_i))$, where $\mathbf{T}_i$ are the factors in $\mathbf{T}$ that depend on $z_i$ and $\bar{\mathbf{T}}_i$ are similarly defined. This shows that $\mathbf{M}$ is a block transformation matrix for each block $[z_i, z_i^2]$ with scaling factor $l_i$. That is, the only possible nonzero element in $\mathbf{M}$ are $M_{i,i}$, $M_{i,2i}$, $M_{2i,i}$, and $M_{2i,2i}$ for all $i \in \{1, \cdots, d\}$.

Furthermore, for any $i \in \{1, \cdots, d\}$ we have that

$$l_i^{-1} = \frac{\partial \mathbf{T}(\mathbf{l}^{-1}(\mathbf{z}))_i}{\partial z_i} = \sum_{j=1}^{k} M_{i,j} \frac{\partial \bar{\mathbf{T}}(\mathbf{z})_j}{\partial z_i} = M_{i,i} + 2M_{i,2i} z_i, \tag{60}$$

$$2l_i^{-1} z_i = \frac{\partial \mathbf{T}(\mathbf{l}^{-1}(\mathbf{z}))_{2i}}{\partial z_i} = \sum_{j=1}^{k} M_{2i,j} \frac{\partial \bar{\mathbf{T}}(\mathbf{z})_j}{\partial z_i} = M_{2i,i} + 2M_{2i,2i} z_i. \tag{61}$$

This implies that $M_{i,2i} = 0$ and $M_{2i,i} = 0$ for any $i \in \{1, \cdots, d\}$, and $M_{i,i} = l_i^{-1}$ for $i \in \{1, \cdots, k\}$, which reduces $\mathbf{M}$ from a block transformation matrix to a permutation and scaling matrix. In particular, this means that the latent factors $z_i$ are identifiable up to permutations and scaling, with the transformation matrix $\mathbf{P} \in \mathbb{R}^{d \times d}$ defined by the first $d$ rows and $d$ columns of $\mathbf{M}$:

$$\mathbf{A}^{-1}\mathbf{h} = \mathbf{P}\mathbf{A}_*^{-1}\mathbf{h} + \mathbf{r} \quad \Longleftrightarrow \quad \mathbf{h} = \mathbf{A}\mathbf{P}(\mathbf{A}_*^{-1}\mathbf{h}) + \mathbf{A}\mathbf{r}. \tag{62}$$

Since $\mathbf{h}$ is linearly identifiable by assumption, it must be that $\mathbf{A}\mathbf{r} = \mathbf{0}$ by Definition 3.1. Since $\mathbf{A}$ is invertible by assumption, it must be that $\mathbf{r} = \mathbf{0}$. Therefore, we have

$$\mathbf{A}^{-1}\mathbf{h} = \mathbf{P}\mathbf{A}_*^{-1}\mathbf{h}. \tag{63}$$

This completes the proof. $\qquad\qquad\qquad\qquad\qquad\qquad\qquad\qquad\qquad\qquad\qquad\qquad\square$

## C  MODEL CONFIGURATIONS

In MTRN, the learnable parameters are the feature extractor parameters $\phi$ and the task-specific regression weights $\mathbf{w}_t$ for all tasks $t$. These model parameters are learned by maximum likelihood as defined in Equation (1).

In MTLCM, the learnable parameters are the linear transformation $\mathbf{A}$, the causal indicators $\mathbf{c}_t$ for all tasks $t$, and the spurious coefficients $\boldsymbol{\gamma}_t$ for all tasks $t$. These are free parameters learned by maximum marginal likelihood as defined in Equation (13). The binary causal indicators $\mathbf{c}_t$ are parameterized as free parameters squashed to $[0, 1]$ by the sigmoid function. To allow for gradient update of $\mathbf{c}_t$, we do not binarize the output of the sigmoid function during training; instead, we use a soft version $\tilde{\mathbf{c}}_t \in [0, 1]^d$ during training. In practice, we find that this works well and all learned values for $c_{t,1}$ are very close to either 0 or 1. In the synthetic data setting, the learned causal indicators match the ground-truth values. In practice, we find that fixing the spurious noise variance $\sigma_s$ to 0.01 and the observational noise variance $\sigma_0$ to 0.1 works well for all experiments.

For a fair comparison, we also consider the multi-task extensions of iVAE and iCaRL, MT-iVAE and MT-iCaRL, which include the task variable $t$ in the conditioning variables $\mathbf{u}$ in their conditional

priors $p_{\mathcal{T}}(\mathbf{z}|\mathbf{u})$, with the task-specific parameter $\mathcal{T}(t) = \{\mathbf{v}_t\}$ to be learned from data, which is the counterpart to $\mathcal{T}(t) = \{\mathbf{c}_t, \boldsymbol{\gamma}_t\}$ in our MTLCM but has no explicit interpretations with respect to a causal graph. We set $\dim(\mathbf{v}_t) = \dim(\mathbf{c}_t) + \dim(\boldsymbol{\gamma}_t)$ to ensure the same degree of flexibility as our MTLCM. The task-specific parameters $\mathbf{v}_t$ are free parameters learned together with other parameters in these models by optimizing their variational/score matching objective.

## D  DETAILS ON THE UNINFORMATIVE PRIOR OVER THE REGRESSION WEIGHTS

Since no prior knowledge is assumed for the task-specific regression weights $\mathbf{w}_t \in \mathbb{R}^d$, we put an uninformative prior over $\mathbf{w}_t \in \mathbb{R}^d$ for all tasks $t$:

$$p(\mathbf{w}_t) \propto 1. \tag{64}$$

Since the support of $\mathbf{w}_t$ is $\mathbb{R}^d$, such an uninformative prior can be thought of as a Gaussian prior with infinite variance whose density is zero almost everywhere.

We marginalize out $\mathbf{w}_t$ from $p_{\mathcal{T}}(y|\mathbf{z}_c, t) = \mathcal{N}(y|(\mathbf{w}_t \circ \mathbf{c}_t)^{\mathrm{T}}\mathbf{z}, \sigma_p^2)$ under the uninformative prior over $\mathbf{w}_t$, which makes the conditional prior over $y$ uninformative:

$$p'_{\mathcal{T}}(y|\mathbf{z}_c, t) = \int p_{\mathcal{T}}(y|\mathbf{z}_c, t)p(\mathbf{w}_t)d\mathbf{w}_t \propto 1. \tag{65}$$

Therefore, we have

$$p_{\mathcal{T}}(\mathbf{z}|y, t) = \frac{p_{\mathcal{T}}(\mathbf{z}_c|t)p'_{\mathcal{T}}(y|\mathbf{z}_c, t)p_{\mathcal{T}}(\mathbf{z}_s|y, t)}{\int p_{\mathcal{T}}(\mathbf{z}_c|t)p'_{\mathcal{T}}(y|\mathbf{z}_c, t)p_{\mathcal{T}}(\mathbf{z}_s|y, t)d\mathbf{z}_s d\mathbf{z}_c} \tag{66}$$

$$= \frac{p_{\mathcal{T}}(\mathbf{z}_c|t)p_{\mathcal{T}}(\mathbf{z}_s|y, t)}{\int p_{\mathcal{T}}(\mathbf{z}_c|t)p_{\mathcal{T}}(\mathbf{z}_s|y, t)d\mathbf{z}_s d\mathbf{z}_c} \tag{67}$$

$$= p_{\mathcal{T}}(\mathbf{z}_c|t)p_{\mathcal{T}}(\mathbf{z}_s|y, t). \tag{68}$$

Since $p_{\mathcal{T}}(\mathbf{z}_c|t)$ factorizes over the causal latent factors and $p_{\mathcal{T}}(\mathbf{z}_s|y, t)$ factorizes over the spurious latent factors, the structured conditional prior $p_{\mathcal{T}}(\mathbf{z}|y, t)$ factorizes over all latent factors $\mathbf{z}$.

## E  EXPERIMENT SETTINGS FOR THE SYNTHETIC DATA

We detail the precise process for the data generation of the synthetic data for both the linear and non-linear experiments below. Algorithm 1 details the full data generation process, Table 4 details the experiment hyperparameters used in the linear setting and Table 5 details the hyperparameters used in the non-linear setting. The transformation in the linear experiments corresponds to either the identity, an orthogonal or a random matrix of size $latent\_dim \times latent\_dim$, while in the non-linear experiments it corresponds to a randomly initialized neural network with the specified hidden dimensions and relu activations.

## F  ABLATION STUDY FOR THE LINEAR SYNTHETIC DATA

In Figure 4, we contrast the effect of training only the linear transformation matrix $\mathbf{A}$ in our MTLCM when the ground-truth task variables $\mathbf{c}_t, \boldsymbol{\gamma}_t$ are known to the model, with the more general setting of learning all parameters jointly via maximum marginal likelihood. We assess the convergence of our multi-task linear causal model across 5 random seeds for increasingly complex linear transformations (identity, orthogonal, random) for data consisting of 10 latent factors with two causal features. Rather than inhibiting convergence, we find that training all parameters jointly leads to improved performance, possibly due to additional flexibility in the parameterizations of the model. For all types of linear transformations, our model succeeds in recovering the ground-truth latent factors.

**Algorithm 1** Pseudocode for the data generating process in the synthetic data experiments

**Require:** $l$ the number of latent features
**Require:** $N_c$ the number of causal features
**Require:** $N_t$ the number of tasks
**Require:** $N_s$ the number of points per task
**Require:** Ground-truth transformation F (random invertible matrix or random MLP)
  Let $\sigma_s, \sigma_o = 0.1, 0.01$
  **for** Each task t **do**
    Sample $l$ binary causal feature indicators $I_1, I_2, \cdots, I_l$
    Sample $l$ weights $w_j^t$ from $\mathcal{U}(0,1)$
    Sample spurious coefficients $\gamma_j$ from $U(-1,1)$ for all $I_j = 0$.
    **for** each data point with index i in this task **do**
      Sample causal features $Z_i^j \sim \mathcal{N}(0, \sigma_s^2)$ for all $I_j = 1$
      Sample $\sigma_p \sim \mathcal{U}(2,3)$
      Obtain target Y $= \sum_{j|I_j=1} Z_i^j + \mathcal{N}(0, \sigma_p^2)$
      Obtain spurious features $Z_i^j = \gamma_j * Y + \mathcal{N}(0, \sigma_s^2)$ for all $I_j = 0$
      Obtain observed features via the transformation $\mathbf{x_i^t} = F(\mathbf{z_i^t}) + \mathcal{N}(\mathbf{0}, \sigma_o^2\mathbf{I})$
    **end for**
  **end for**

Table 4: Experimental Settings for the Linear Synthetic Data

| Latent Dim | 3, 5, 10, 20, 50, 100 |
|---|---|
| Num Causal | 2, 4 |
| Seed | 1, 2, 3, 4, 5 |
| Matrix Type | random |

Table 5: Experimental Settings for the Non-Linear Synthetic Data

| Observation Dim | 50, 100, 200 |
|---|---|
| Encoder Network Num Hidden | 1 |
| Encoder Network Hidden Dim | 2 * Observation dim |
| Latent Dim | Observation dim |
| Num Causal | 4, 8, 12 |
| Seed | 1, 2, 3, 4, 5 |

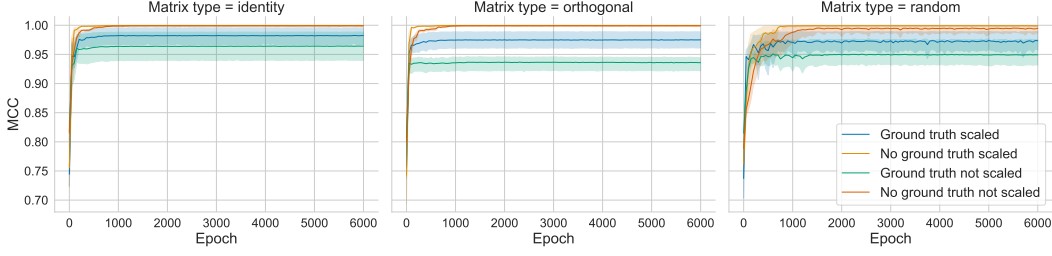

Figure 4: Convergence of the model in the case of transformations of the latent factors for identity, orthogonal and arbitrary linear transformations. Standardizing the features accelerates convergence.

