# OpenReview forum: "Leveraging Task Structures for Improved Identifiability in Neural Network Representations"
_ICLR.cc/2024/Conference — Submitted to ICLR 2024_

### Official Review · Reviewer_wjky · 2023-11-01

**Soundness:** 2 fair
**Presentation:** 4 excellent
**Contribution:** 3 good
**Rating:** 6
**Confidence:** 3

**Summary:**

The paper studies the problem of identifying causal representations in the multi-task regression setting. The considered regression function consists of an invariant feature extractor and task-dependent linear head. The identifiability of the feature extractor and linear head is provided (up to some equivalent classes). The main concern for the work is the model assumption on the relation between the target variable and spurious features.

**Strengths:**

1. The paper is well-written overall.

2. Solid identifiability results are provided.

**Weaknesses:**

1. In the assumed causal graph, $Y$ is a parent of the spurious features $z_{s}$, which is not a natural assumption. The correlation between $Y$ and $z_{s}$ is often due to latent confounders.

2. What does ``MLE (1) converges" in Corollary 3.3 mean? $\theta'$ defined in (1) is a population estimator. There is a similar issue for  (17) in Theorem 3.5. Besides, the uniqueness of the (population) MLE estimators (i.e., (1) and (17)) should be assumed explicitly.

3. What is $c_{t}$ below equation (4)?

**Questions:**

The main concern is the causal relation between $Y$ and the spurious features $z_{s}$. If there are indeed many real-world settings where $Y$ is the parent for the spurious features. Please cite related references to support the assumption. Otherwise, the identifiability for settings with the latent confounder should be studied as well.

I may raise my score based on the response to this question.

---

> ### Author Response · Authors · 2023-11-17
>
> Thank you for taking the time to review our work. Regarding the primary concern of the target relationship with the spurious variables, we can first begin by acknowledging that indeed this assumption does not in general capture all possible non-causal correlations between latent features and $Y$. The Reichenbach principle states that such correlations (when non-causal) must either originate from a common cause (confounders) or from an anti-causal relationship, as assumed in this work.
>
> When dealing with models over unobserved causal variables which are non-linear transformations of the input space, it can be difficult to provide intuition for real-world settings. However, this anti-causal relationship (as in the paper) is well-documented in real-world examples in epidemiology. See Figure 1 in [1], treating perceived pandemic impact or IES-R score as the regression target, or Figure 6 (right) in [2], where testing status may well be included as a feature in estimating Case Fatality Rates, but there is likely to be causal influence between overall case fatality rate and testing policy. Broadly, any form of selection bias may lead to similar cases, where the selection criterion may be included in the feature set. A similar example can be imagined in most drug discovery campaigns, where molecules to be tested are selected based on some structural similarities to an originally promising molecule (based on the quantity to be estimated, e.g. drug potency). Structural molecule features are then likely to be spuriously correlated with the regression target due to their selection criteria, without actually being involved in the drug’s mechanism of action.
> Thus the first portion of our answer is that there are many situations where the proposed model may be useful in practice, even if it does not explicitly model the confounders in full generality.
>
> The second portion of our response is that any confounding variable on the spurious variable $z_s$ would also be a causal variable for $Y$, and thus already be included in the set of latents being modelled. Since this work is concerned with identifying the correct latent variables as opposed to the full connectivity of the underlying causal graph (which is a more general but much more challenging problem), the existence of an additional causal relation from a causal latent variable to a “spurious” one is thus possible but does not factor into our model.
>
> ### Other Points
>
> For point 2, we have added the explicit uniqueness assumption for the MLE estimator and reworded “convergence” to more precise terminology. Convergence here refers to both convergence of the optimization procedure together with standard regularity conditions for consistency of MLE estimators.
>
> For point 3, $c_t$ refers to the indices of the causal variables for task t. This is explained in the paragraph preceding equation (4):
>
> > Let $\mathcal{T}(t)=\{\mathbf{c}_t,\boldsymbol{\gamma}_t\}$ be a collection of task-specific variables associated with task $t$, which are free parameters to be learned from data, where $\mathbf{c}_t\in\{0,1\}^d$ are the causal indicator variables which determine the partition of $\boldsymbol{z}=\boldsymbol{z}_c \cup \mathbf{z}_s$ for the given task $t$ (i.e $c_t^i=1$ indicates that $z_i$ is a causal latent factor in task $t$ and $c_t^i=0$ indicates that $z_i$ is a spurious latent factor in task $t$).
>
> We hope that this has sufficiently addressed your concerns.
>
> [1] Wang, Cuiyan, Agata Chudzicka-Czupała, Michael L. Tee, María Inmaculada López Núñez, Connor Tripp, Mohammad A. Fardin, Hina A. Habib, et al. 2021. “A Chain Mediation Model on COVID-19 Symptoms and Mental Health Outcomes in Americans, Asians and Europeans.” Scientific Reports 11 (1): 6481. https://doi.org/10.1038/s41598-021-85943-7.
>
> [2] “Simpson’s Paradox in COVID-19 Case Fatality Rates: A Mediation Analysis of Age-Related Causal Effects.” 2021. Ieee Transactions on Artificial Intelligence 2 (1): 18–27. https://doi.org/10.1109/TAI.2021.3073088.

---

> > ### Comment · Reviewer_wjky · 2023-11-18
> > **reply**
> >
> > 1. In discussions of spurious correlation, the main context is confounding, since there is often no well-understood causal relations between certain features and the target. From a theoretical perspective, I think it is of interest to study the considered setting. Besides, the authors provide some real-world examples for the considered setting. Therefore, I increase my score.
> >
> > 2. Thanks for the clarification of $c_{t}$.

---

### Official Review · Reviewer_mdpZ · 2023-11-02

**Soundness:** 3 good
**Presentation:** 2 fair
**Contribution:** 2 fair
**Rating:** 5
**Confidence:** 2

**Summary:**

The authors propose an identification result under the hypothesis that data are generated by a specific graphical model.

**Strengths:**

The identifiability result seems new

**Weaknesses:**

My main complaint is the clarity of the paper.
I am not sure I properly understood the setup or the contribution: no algorithm is clearly proposed.

- Abstract "In such cases, we show that identifiability is achievable even in the case of regression, extending prior work restricted to linear identifiability in the single-task classification case." I am not sure to understand this sentence since the authors refer to previous identification results in the multitasks setting (section 3.2 for instance)

- Section 3.3.1 is a succession of 6 equations, is it possible to encapsulate the assumptions in a proper environment

**Questions:**

- What's the proposed algorithm? I was not able to parse it.

---

> ### Author Response · Authors · 2023-11-17
>
> Thank you for your comments. See below our point by point response:
>
> - This sentence in the abstract refers to the extension of Roeder et al’s work on supervised learning identifiability. Given that our linear identifiability result most closely resembles this work, and that we view Lachapelle et al’s work as concurrent to ours, this phrasing was used. Nonetheless, we have reworded this in the abstract to avoid any confusion. The sentence now reads: “In such cases, we show that linear identifiability is achievable in the general multi-task regression setting. Furthermore, …”
>
> - The presentation of 3.3.1 was chosen to make clear, with textual explanation, where each equation comes from. Together they define our model. Since most reviewers appreciated the presentation of our paper, we are hesitant to change this. Please see below our condensed explanation of the method and let us know if this is still unclear. We are happy to elaborate further.
>
> The overall method is a two stage algorithm. This algorithm applies to the setting of multi-task regression, i.e. when multiple continuous outputs are associated to a joint set of input points. This is often relevant for example in drug discovery, where one would like to jointly model various relevant properties of a drug candidate. The first stage of the algorithm is simply to train a standard multitask regression neural network (which has separate linear heads $w_t$ for each task and a shared feature extractor $f(x)$ across all tasks) jointly on all tasks. Theorem 3.2 says that the resulting last-layer representation of the neural network will be linearly identifiable. The second stage of our algorithm starts from this linearly identifiable representation and “disentangles” it via our probabilistic model. Here, we recover the linear transformation $A$ (see equation 10) by maximizing the likelihood obtained in equations 11 and 12. Note that this general procedure is summarized in the manuscript in the last paragraph of section 3.1, and illustrated in Figure 2. We are again hesitant to change the presentation here given the generally positive feedback in this regard, but are happy to clarify further.
>
> Given that the reviewer’s only concern appears to be the clarity of the work, which has otherwise been well received, we would kindly urge them to reconsider their rating as we do not believe this warrants a rejection score. If there are still specific elements which are unclear that the reviewer would like to point out, we are also happy to clarify them in the manuscript.

---

### Official Review · Reviewer_NHKJ · 2023-11-03

**Soundness:** 2 fair
**Presentation:** 3 good
**Contribution:** 2 fair
**Rating:** 5
**Confidence:** 3

**Summary:**

This work studies causal representation learning for multi-task regression data.  The authors assume a data generating process under which

1. the response variable $y$ is always a linear function of the causal and spurious latent variables $z$, with the linear map varying across tasks,
2. the observed features $x$ are informative about the latent variables and the tasks are sufficiently diverse,
3. the conditional distributions $p(y\mid x, t) = N(y\mid w_t^\top f(x), \sigma_t^2 I)$ and $p(f(x)\mid z)$ are homoscedastic Gaussian,

and propose a two-stage procedure that can recover the causal latent variables up to permutations and scaling.  The method outperforms previous approaches on synthetic data, and is demonstrated to produce stable outputs on two real-world datasets.

**Strengths:**

- The authors studied an important problem.
- The proposed method demonstrates promising empirical performance.
- Compared to recent works on the same problem, the proposed method is more computationally efficient.

**Weaknesses:**

1. My main concern is the Gaussianity assumptions made throughout the work.  Such assumptions rarely hold in practice; and in contrast to standard machine learning tasks where most methods remain useful even if such assumptions are violated (e.g. typical regression procedures continue to estimate the conditional expectation), for causal representation learning the utility of the proposed method is far less clear.

2. The above concern mostly applies to the second stage of the procedure.  For the first stage it appears that the method only relies on the conditional expectation having the form of $E(y\mid x) = w_t^\top f(x)$ (plus bijectivity and task diversity), but in such cases I feel that the assumption is better stated just as in Lachapelle et al (2023), namely we can learn a feature extractor that is linearly equivalent to the ground truth.

3. Regarding experiments, I think it would be more convincing if the authors could include comparisons to the recent works of Lachapelle et al (2023) and Fumero et al (2023), as they addressed the same problem and have been publicly available for 4-5 months.

4. Finally, I find the feature-space linearity assumption $E(y\mid x) = w_t^\top f(x)$ somewhat unsettling for the use in causal representation learning.  While similar assumptions have appeared in stylized theoretical analyses for multi-task learning, designing a causal representation learning procedure based on such assumptions appears to be asking much more.

**Questions:**

See above, in particular points 1-3.

---

**Post-rebuttal update.** Thank you for your response.  Unfortunately, I remain concerned about the assumptions and presentation of the results, and I will keep my score unchanged.

- I do not agree that the feature space linearity is "not an assumption": this condition, and theorem 3.2, only hold because of the conditions on the data generating process introduced in Section 3.1.   Your claim "any non-linear neural network broadly makes this assumption" is somewhat misleading because for general task distributions (on which the conditions in Section 3.1 may not hold *for any fixed latent dimensionalities*), for the condition to hold approximately we may need the NN feature dimensionality to grow w.r.t. the number of tasks, thereby violating the rank condition in Theorem 3.2.

- I am not convinced that the assumption of a *correctly specified* Gaussian likelihood is not a natural choice, as is also noted b reviewer 2cPA.  Currently your proofs rely on the likelihood being correctly specified; the best way to address this concern is to rewrite them so that they apply to (certain) misspecified cases.

---

> ### Author Response · Authors · 2023-11-17
>
> We thank the reviewer for taking the time to review our work. First, we appreciate the recognition that this work addresses an important problem, namely that of defining models which have more robust guarantees on the types of representations they learn.
>
> We would like to emphasize that the probabilistic assumptions formulated in our model not only make our model more computationally efficient, they also enable direct optimization of the underlying likelihood. This is in contrast to iVAE (which optimizes a variational objective) and iCaRL (which optimizes the score function), and the work of Lachapelle et al (2023), which proposes a bi-level stochastic optimization procedure. This is a key property which leads to the stability of our empirical results. We also believe that our formulation most plainly establishes the connections between identifiability and multi-task learning. We thus believe that the reviewer is undervaluing the contributions of our work.
>
> The reviewer’s main concern lies in our Gaussianity assumptions. We feel that points 1 and 4 may be related in this regard:
>
> - The feature space linearity (point 4) is accurate precisely because of the first stage of our procedure: namely this is not an assumption but a guarantee which follows from linear identifiability of the representations learned from the multitask regression network (theorem 3.2). More generally, any non-linear neural network broadly makes this assumption (where the conditional expectation is parametrized by a linear combination of the last layer activations), which is in general less well motivated than in this work given the lack of generic identifiability results. We would be happy to clarify this further if we are misunderstanding any of the reviewer’s concerns here.
>
> - Because of the feature space linearity, the second stage procedure reduces to a linear problem. A gaussian likelihood is the widely accepted standard for linear regression models, and is thus a very natural choice. Given that the linearity only arises in the context of arbitrarily non-linear feature mappings $f(x)$ provided by the multitask neural network in the first stage, we do not believe this to be particularly restrictive. We would point to our empirical evaluation as evidence that our model can indeed be useful. The connection to our causal representation learning result, recovering the linear transformation to the latents, is simply a very appealing byproduct of maximizing the likelihood in equation (11); it does not inherently require different assumptions on the regression setting.
>
> Regarding point 3, we agree that these are relevant papers to our work. Fumero et al primarily focuses on the meta-learning setting and does not provide identifiability results, however we agree that the work by Lachapelle et al would be a valuable baseline as it provides a more detailed discussion of identifiability. We will be incorporating this baseline into our experiments and update the paper accordingly in the coming days, however we urge the reviewer to reconsider our contributions even without these additional results.

---

> ### Author Response · Authors · 2023-11-21
> **Comment on additional baseline**
>
> Please see above our [general response](https://openreview.net/forum?id=kkQSwtx0p3&noteId=S2z6YEyE1r) regarding the requested baseline experiments. We look forward to clarifying any further concerns.

---

### Official Review · Reviewer_2cPA · 2023-11-14

**Soundness:** 3 good
**Presentation:** 3 good
**Contribution:** 2 fair
**Rating:** 5
**Confidence:** 4

**Summary:**

This paper shows that the latent variables of a multi-task regression network can be identified up to an affine transformation. By further assuming a specific causal structure and exploiting changes in the set of variables which are causal vs. spurious, one can identify the latent causal variables more strongly, up to scaling and permutations. Authors illustrate the benefits of this type of identifiable causal representation learning on synthetic and real-world data.

**Strengths:**

1. With some exceptions discussed below, the paper's main parts are clearly written and in general it's nice and simple presentation

2. Working with Gaussian has allowed the authors to create a nice MLE algorithm that avoids the complications and computational cost of many other approaches

3. Empirical results seem fairly broad and convincing, with the exception of some questions below

**Weaknesses:**

1. **Main weakness is that it feels like this paper makes very specific assumption that are often unrealistic and restrictive**, for example it feels very restrictive and unrealistic to assume that the causal latents are just zero-mean standard Gaussians and only the indentity of the causal/spurious variables changes ($c_t$). The spurious factor generation has similar issue with being not so general. Yet another assumptions is that of uninformative prior on the weights (but none on the covariances), which feels more like a technical trick that is required. From causal perspective, it feels limiting to assume a linear form for generating $y$ It is thus hard to see what is a naturally suitable application are for this model. Finally, assuming that there is a certain number of latent variables which then switch across tasks on whether they are spurious/causal seems again simplistic (but would love to hear if I'm wrong on this!).

2. **Some of the claims with respect to identifiability results feel exaggerated**: "*We emphasize that this is a stronger identifiability result than identifiability up to block permutations and scaling (i.e., strong identifiability) as in prior works*", "*We emphasize that this strictly strong identifiability class is stronger than the strong identifiability class as in prior works (Khemakhem et al., 2020a; Lu et al., 2022) which are only identifiable up to*" -- Similarly strong identifiability results have been shown in a general setting in Halva et al. (2021), please see their Theorem 2 in particular and its proof that share similar ideas. Theorem 3, 4 in Halva and Hyvarinen (2020) show strong identifiability in a more specific case, with the similarity here being the exploitation of linear independence and Gaussianity. The authors have cited these works elsewhere and should therefore be also aware of their identifiability results. Further, Morioka et al. (Independent Innovation Analysis, 2021) shows similar identifiability results and proof in their Theorem 2 (though for a different model). Theorem 3.2. and the MTRN identifiability results appear to be essentially the same as those of iVAE (Khemakhem et al.) so I am having hard time understanding whether it has any novel contribution in itself or whether it only sets up the 'second part' with stronger identifiability. Further, the

Other issues:
- Would be nice to have some references/citations if you make strong statements like this "This contrasts with current state-of-the-art approaches, whose assumptions also fit our assumed data generating process but which are difficult to train effectively and only identifiable up to block permutations and scaling." Which works are these? Why are they difficult to train, and where is that shown? And again last part about identifiability is not even true as shown above.
- "... Willetts & Paige (2021); Kivva et al. (2022) recently extend the results in unsupervised generative models to the case of models with mixture model priors." Not all these papers assume mixture model priors specifically
- Given the following statement, you should have included them as baselines: "concurrent work (Lachapelle et al., 2023; Fumero et al., 2023) has expanded this area of research to consider the multi-task and meta-learning settings."
- Figure 2 appearing before Figure 1 is always confusing...
- "...existence of d independent ground-truth..." please be careful of the word independent and how it's used
- "The marginal likelihood of $\psi$ under MTLCM is given by" imprecise language -- it's not the likelihood of $\psi$ as they are not random variables.

**Questions:**

1. Both $\mathbf{x}$ and $y$ appear to be deterministic functions of $\mathbf{z}$, at the same time you seem to assume a noise model e.g. for $p(y\mid x, t)$ has normal distribution with covariance term $\sigma_{r, t}^2$. What is the source of this covariance -- please explain whether it's due the 'push-forward' of $\mathbf{z}$ through the deterministic functions or whether there is some output noise on top of that. Also what is that subscript $r$ on that variance term?

2. Please comment what is the difference between the iVAE identifiability results and the ones here shown form multi-task regression -- they seem very similar.

3. In experiments: Why do you only consider 1 layer MLP for nonlinearity? What if there are more layers? Could you also explain this in more detail: "we find that the strong MCC for our MTLCM is able to match the weak MCC for the MTRN." as well as what you exactly mean by this "MCC score between the data representations recovered by each pair of those 5 models,", and also this in more depth ".. weak identifiability achieved from the MTRN implies that identifiability is achievable up to eight latent features, suggesting there may be some redundancies between tasks"

---

> ### Author Response · Authors · 2023-11-17
>
> Thank you for the insightful review. We appreciate that you recognize the general strengths of our work. Taken together, we believe these strengths constitute a contribution which is better than fair to the field. The points raised are valid and can for the most part be readily addressed. See our point by point response with the questions addressed in the comment below.
>
> ### Main Concern
>
> * The standard gaussian assumption for the causal latents is only a prior, and a very common choice. We note that one could choose any other isotropic gaussian prior without significantly changing the derivation, but we do not see a reason to do so. It also seems reasonable to not assume additional covariance structure a priori.
>
> * The linearity of the causal model, and the way the causal structure varies, appears to be the main concern. Regarding linearity, we can only note that this is only with respect to the latent feature space which is learned in the first stage of our proposed procedure, by an arbitrarily complex neural network $f(x)$. This mirrors the way most regressors are generated and is further validated by our linear identifiability result.
>
> * On how the latent structure varies across tasks: this is a core assumption of our model and it underpins our hypothesis for *why and when* multi-task learning is a useful thing to do, namely that there is shared causal structure between tasks. One way to see this is that it is a sufficient, but not necessary, condition for the usefulness of multi-task learning. Nonetheless, we do not think this assumption is as restrictive as the reviewer seems to believe. Consider the alternatives, under the condition that the overall number of latent variables is sufficiently large:
>
>     i. A given latent variable could have no correlation with the regression target Y for a given task t. In this case, the model would still be able to parameterize this by treating this latent as a causal variable with regression weight 0.
>
>     ii. A given latent variable could have a common cause parent with Y. As was stated in the response to reviewer NHJK, this common parent would then itself be a causal latent variable. While we do not capture the direct relationship between the causal $z_c$ and spurious $z_s$ variables in this case, we are concerned with the identifiability of the latents as opposed to a full causal graph.
>
> ### Other Points
>
> 1. On the statement contrast with current SOTA approaches, this statement referred to iVAE and iCaRL, the supporting evidence for which is in our experiments.  We have clarified this in the paper. Furthermore, the Lachapelle et al (2023) approach is also difficult to train due to its bilevel optimization objective.
> 2. On the citation for identifiable unsupervised learning with mixture model priors, we have removed the two references which were not applicable for this statement.
> 3. We are working to include the Lachapelle et al (2023) approach in our baselines. The Fumero et al 2023 paper does not propose identifiability results, and is less relevant since it focuses on meta-learning rather than multitask learning.
> 4. “..existence of d independent…” has been reworded to “existence of d linearly independent ground-truth task-specific vectors..”
> The statement on marginal likelihood has been corrected.

---

> ### Author Response · Authors · 2023-11-17
>
> ### Questions
>
> 1. $Y$ is not a deterministic function of $z$ in our setting because we assumed Gaussian noise for their relationship as shown in Equations (5) and (6). Therefore, $p(y|x,t)$ is also non-deterministic (Gaussian). In general, it is common to assume randomness in $p(y|x,t)$ due to, e.g., observation noise. The subscript $r$ (r for regression) distinguishes $\sigma_{r,t}$ from other noise variance variables such as $\sigma_o$, $\sigma_p$ and $\sigma_s$ for the linear model in the second stage.
>
> 2. The difference is that iVAE is only identifiable up to block permutations and scaling of the sufficient statistics whereas our Theorem 3.5 extends that to permutations and scaling of the actual latent variables by leveraging the properties of the linear likelihood as defined in Equation (10). Specifically, our novel contribution lies in Equations (60)-(63) which are not possible in the iVAE setting.
>
> 3. The synthetic experiment with nonlinear data is just a proof of concept which verifies that our multitask regression networks (MTRNs) can produce linearly identifiable representations from nonlinearly transformed inputs, and we believe that one layer of MLP nonlinearity is sufficient for this purpose. We have experimented with more MLP layers and the results remain the same.
>
> Below we clarify the sentences that the reviewer asked.
>
> > …the strong MCC for our MTLCM is able to match the weak MCC for the MTRN.
>
> This means that the MTLCM successfully produces permutation-identifiable latent variables from the linearly identifiable latent variables produced by MTRN. In theory, the weak MCC for the MTRN is an upper bound for the strong MCC for MTLCM, because the MTLCM is trained on the representations produced by MTRN. If they match (i.e., they are approximately equal), it would imply that MTLCM is fully permutation-identifiable.
>
> > …MCC score between the data representations recovered by each pair of those 5 models
>
> For real-world data, we do not have access to ground-truth latent variables, so we cannot evaluate our model by computing the MCC between the latent variables recovered by our model and the ground truth. In such case, it is common (e.g., see Khemakhem et al. 2020b) to train a few models from different random initializations with different random seeds and see whether the latent variables recovered by these models are identifiable against each other (i.e., whether the MCC score between the latent variables recovered by each pair of these models are high). If so, this would imply that they always find solutions within the same equivalence class, and hence the model is identifiable up to that equivalence class.
>
> > …weak identifiability achieved from the MTRN implies that identifiability is achievable up to eight latent features, suggesting there may be some redundancies between tasks.
>
> By redundancies, we mean that there exist tasks in QM9 which have linearly dependent ground-truth weight vectors. In Corollary 3.3, one of the assumptions for MTRN to be linearly identifiable is that the number of tasks with linearly independent ground-truth weight vectors should be greater than or equal to the number of latent variables used in the MTRN model. For this task (QM9), we found that the weak MCC score was low if we used more than eight latent variables. This implies that there are eight tasks with linearly independent ground-truth weight vectors.

---

> > ### Comment · Reviewer_2cPA · 2023-11-23
> >
> > I have looked at the comments and the revised paper but will maintain my point that there is not a just acknowledgment of previous results. For example, works I already mentioned in my earlier comments such as [1], [2] already achieve equally strong identifiability results and the authors have not covered that. Also as mentioned, Section 3.2 on MTRN and Theorem 3.2. (i.e. stage 1) appears to be just recycling essentially the same results as those in iVAE (Khemakhem et al.). I don't feel like the work has enough theoretical novelty.
> >
> > [1] Morioka et al. (2020, independent innovation analysis)
> > [2] Halva et al. (2021, disentangling identifiable features)

---

> ### Author Response · Authors · 2023-11-21
> **Comment on additional baseline**
>
> Please see above our [general response](https://openreview.net/forum?id=kkQSwtx0p3&noteId=S2z6YEyE1r) regarding the requested baseline experiments. We look forward to clarifying any further concerns.

---

> ### Author Response · Authors · 2023-11-23
> **Clarification on differences with previous results**
>
> Thank you for your response. We respectfully disagree about the statements that [1] and [2] provide equally strong identifiability results. There are multiple differences between these works and ours. In both of these cases, and with iVAE, the nature of the permutation identifiability is weaker than ours. Specifically, each of these prior works are permutation-identifiable either up to pointwise non-linearities of the latent components [1] and [2], or up to block-permutations of the sufficient statistics, not of the latent variables themselves (iVAE). See the specific passages in each prior work:
>
> [1] : “We show that IIA guarantees the identifiability of the innovations with arbitrary nonlinearities, up to a permutation and componentwise invertible nonlinearities”
>
> [2]: “$f^{−1}$ can be recovered up to permutation and coordinate-wise transformations from the distribution of $(f(s_{t_1} ), . . . ,f(s_{t_{m}}))$” (Theorem 2). “Coordinatewise” here has no restrictions and thus implies non-linearities as well.
>
> iVAE: “. That is, we can recover the original latent variables up to a component-wise (pointwise) transformations $T_i^{*}$, $T_i$, which are defined as the sufficient statistics of exponential families.”
>
> And
>
>  “These two Theorems imply that in most cases $f^{-1} \circ f : Z → Z$ is a pointwise nonlinearity, which essentially means that the estimated latent variables $z$ are equal to a permutation and a pointwise nonlinearity of the original latents $z$.”
>
> Our work provides identifiability up to permutation and scaling (linear) of the actual latent components.
>
> ### Other key differences
>
> 1. All of these methods are optimised using variational inference. This makes our model much simpler to optimise. This is a crucial difference for practical use.
> 2. [2] exploits specific temporal or spatial structure in the encoded latents, while factorising the joint into a switching linear dynamical system, which is a different setting to ours.
>
> Regarding the theoretical novelty of our identifiability results against iVAE, we note that our Theorem 3.2 for MTRN in Section 3.2 is the linear identifiability result for the multi-task regression network, which is completely different to iVAE. We believe that the reviewer meant the similarity between our Theorem 3.5 for MTLCM in Section 3.3.4 and iVAE. Below, we reproduce specific key equations (Eqs 60 and 61 in our paper) that expand upon the identifiability results of iVAE:
>
> $$l_i^{-1}=\frac{\partial T_i (l^{-1}(z))}{\partial z_i}=\sum_{j=1}^k M_{i,j} \frac{\partial \bar{T_j} (z)}{z_i}=M_{i,i}+2M_{i,2i}z_i$$
>
> $$2l_i^{-1}z_i=\frac{\partial T_{2i}(l^{-1}(z))}{\partial z_i}=\sum_{j=1}^k M_{2i,j} \frac{\partial \bar{T_j} (z)}{z_i}=M_{2i,i}+2M_{2i,2i}
> z_i$$
>
> both of which hold for $i=1,\cdots, d$ since $l^{-1}(z)=[l_1^{-1}(z_1),\cdots,l_d^{-1}(z_d)]$ is a linear pointwise function (rather than nonlinear point-wise as in iVAE) thanks to our linear likelihood (this does not hold in iVAEs because their mixing function f is nonlinear). By matching the coefficients, this implies that $M_{i,2i}=0$ and $M_{2i,i}=0$, which reduces the identifiability class from block-identifiable to permutation-identifiable.
>
> We have clarified these key differences in the manuscript, and sincerely hope we have addressed the concerns in this regard

---

### Author Response · Authors · 2023-11-21

To all reviewers, we have addressed each reviewer’s concerns separately below their respective review. We also have updated our manuscript, where changes are highlighted in blue.

Some reviewers have asked that we additionally compare to the work by Lachapelle et al on sparsity and disentanglement. While actively working on this during the rebuttal period, it has come to our attention that this baseline, similar to Fumero et al., also effectively implements a meta-learning setting as opposed to a multitask setting, i.e. it requires that the support and query sets used in the bi-level optimization be disjoint. We hypothesise this may be because when this is not the case, optimising the linear weights in the inner level optimization will lead to near zero gradients in the outer level optimization. Furthermore, the synthetic experiments of this paper require generating 160000 random tasks on the fly during training, and is thus very unlikely to work on real-world data where the number of tasks is significantly smaller (on the order of 12 in QM9, for example).

This latter statement is supported by the theoretical assumptions made in their paper, which requires “Sufficient variability of the task supports - Assumption 3.6”, i.e. that for *every* feature $j$ in the set of latents, there is a set of tasks for which the useful latents cover all features except $j$. As the authors state, this is likely to hold only if the set of tasks (and their supports/useful features) is very large.

For these reasons, in addition to requiring non-trivial modifications of the original baseline code, we do not consider this baseline to be particularly suited to our identifiability experiments and would not expect it to perform well in our setting. We view it primarily as a meta-learning procedure due to the bi-level optimisation, and thus hope the reviewers would agree that the current baselines (iVAE and iCaRL) that we already compare with are more relevant to this work.

Further, while we agree that the Lachapelle et al. work provides valuable insights into multi-task learning and identifiability, we believe these points emphasise the fact that our work’s assumptions may in fact be less restrictive for practical uses than prior work, while offering new perspectives on how this identifiability arises. We hope that this will encourage reviewers to engage with our responses and reconsider their ratings.  If there are any outstanding concerns or clarifications to be made, we would be very glad to clarify them.

---

### Meta-Review · Area_Chair_KW83 · 2023-12-08

**Metareview:**

In the reviews, many critical points have been raised, such as limited theoretical novelty compared to similar papers, highly restrictive (and  possibly non-realistic) assumptions, unclear description of the motivation and the method itself. Most of these concerns could not be fully addressed during the rebuttal and discussion phase. I also share several concerns about the limited conceptual novelty.  Finally, nobody wanted to assign a clearly positive score, and therefore I recommend rejection.

**Justification For Why Not Higher Score:**

Limited conceptual novelty compared to previous papers addressing the same problem.

**Justification For Why Not Lower Score:**

N/A

---

### Decision · Program_Chairs · 2024-01-16

Reject